# Hydrophobic residues in S1 modulate enzymatic function and voltage sensing in voltage-sensing phosphatase

Vamseedhar Rayaprolu[1], Heini M. Miettinen[1], William D. Baker[1], Victoria C. Young[2], Matthew Fisher[1], Gwendolyn Mueller[1], William O. Rankin[1], John T. Kelley[1], William J. Ratzan[1], Lee Min Leong[3], Joshua A. Davisson[1], Bradley J. Baker[3], and Susy C. Kohout[2]

**The voltage-sensing domain (VSD) is a four-helix modular protein domain that converts electrical signals into conformational changes, leading to open pores and active enzymes. In most voltage-sensing proteins, the VSDs do not interact with one another, and the S1–S3 helices are considered mainly scaffolding, except in the voltage-sensing phosphatase (VSP) and the proton channel (Hv). To investigate its contribution to VSP function, we mutated four hydrophobic amino acids in S1 to alanine (F127, I131, I134, and L137), individually or in combination. Most of these mutations shifted the voltage dependence of activity to higher voltages; however, not all substrate reactions were the same. The kinetics of enzymatic activity were also altered, with some mutations significantly slowing down dephosphorylation. The voltage dependence of VSD motions was consistently shifted to lower voltages and indicated a second voltage-dependent motion. Additionally, none of the mutations broke the VSP dimer, indicating that the S1 impact could stem from intra- and/or intersubunit interactions. Lastly, when the same mutations were introduced into a genetically encoded voltage indicator, they dramatically altered the optical readings, making some of the kinetics faster and shifting the voltage dependence. These results indicate that the S1 helix in VSP plays a critical role in tuning the enzyme's conformational response to membrane potential transients and influencing the function of the VSD.**

## Introduction

Voltage-sensing domains (VSDs) are well-known modular domains that cause proteins to be regulated by changes in membrane potential (Hille, 2001). They are composed of four transmembrane helices, labeled S1–S4, with S4 conferring the voltage dependence of the domain via arginines embedded in the helix. The voltage-regulated family of proteins was long thought to be composed solely of channels, but the discovery of the voltage-sensing phosphatase (VSP) broadened the biological role this group of proteins plays in cellular physiology. Instead of having a channel function, VSP connects a VSD to a phosphatase domain that will dephosphorylate phosphatidylinositol phosphates (PIPs) in a voltage-dependent manner (Murata et al., 2005) (Fig. 1 A). Several studies show that the VSD from VSP

functions in the same way as those from channels with S4 movements propagating to the connected linker (Kohout et al., 2008; Villalba-Galea et al., 2008; Murata et al., 2005). The VSP linker then propagates the conformational change to the phosphatase domain, which then removes both the 3- and 5-phosphates from the PIP inositol ring (Murata et al., 2005; Iwasaki et al., 2008; Halaszovich et al. 2009; Kohout et al., 2010; Kurokawa et al., 2012; Castle et al. 2015; Mizutani et al., 2022) (Fig. 1 B). The phosphatase domain is composed of a catalytic domain and a membrane-binding C2 domain with mutations in either domain impacting catalytic activity (Murata et al., 2005; Castle et al. 2015; Paixao et al., 2023). In particular, VSP removes the 5-phosphate from phosphatidylinositol-3,4,5-trisphosphate

[1]Department of Cell Biology and Neuroscience, Department of Microbiology and Cell Biology, Montana State University, Bozeman, MT, USA; [2]Department of Biomedical Sciences, Cooper Medical School of Rowan University, Camden, NJ, USA; [3]Division of Bio-Medical Science and Technology, KIST School, Brain Science Institute, Korea Institute of Science and Technology (KIST), Korea University of Science and Technology (UST), Seoul, South Korea.

Correspondence to Susy C. Kohout: kohout@rowan.edu

V. Rayaprolu's current affiliation is Pacific Northwest Cryo-EM Center, Oregon Health and Sciences University, Portland, OR, USA.   W.D. Baker's current affiliation is School of Medicine, Yale University, New Haven, CT, USA.   M. Fisher's current affiliation is Department of Chemistry and Biochemistry, Montana State University, Bozeman, MT, USA.   W.O. Rankin's current affiliation is Department of Hematology-Oncology, Northwestern University, Evanston, Il, USA.   J.T. Kelley's current affiliation is Revvity Omics, Waltham, MA, USA.   W.J. Ratzan's current affiliation is Department of Systems Biology, Harvard University, Boston, MA, USA.   L.M. Leong's current affiliation is Department of Biological Science, Florida State University, Tallahassee, FL, USA.   J.A. Davisson's current affiliation is Department of Biochemistry, Molecular Biology and Biophysics, University of Minnesota, Minneapolis, MN, USA.

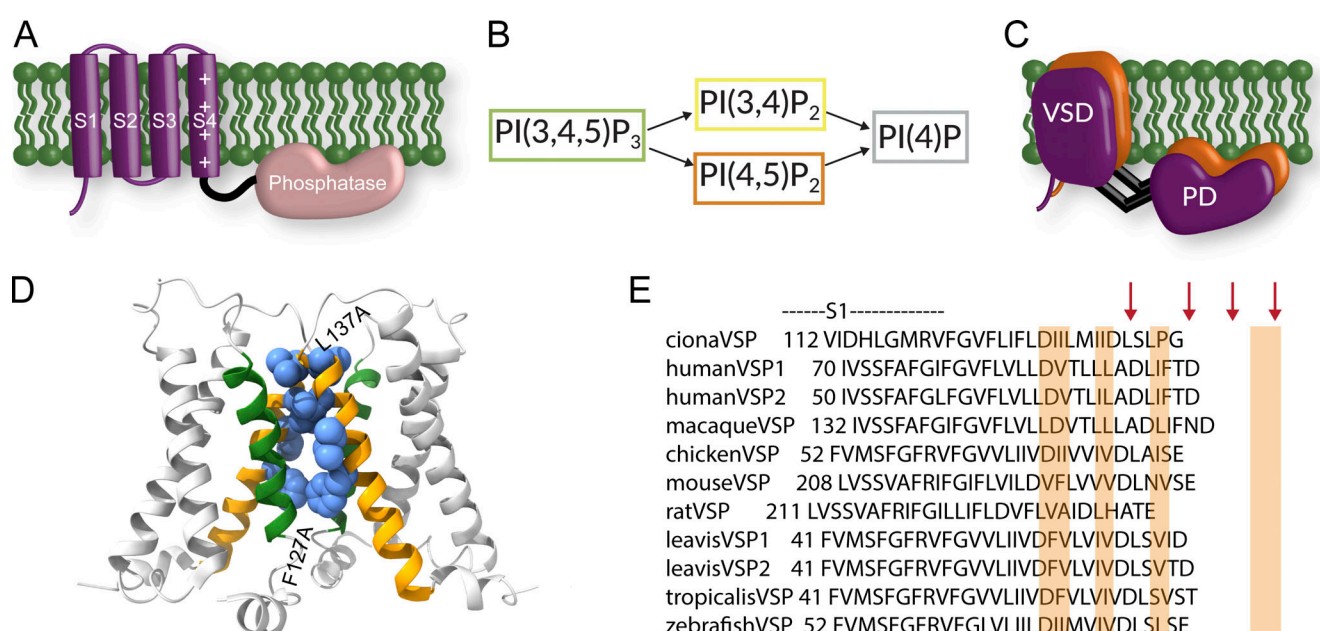

Figure 1. **Conserved S1 helices across VSPs from different species. (A)** Cartoon representation of a single VSP subunit with the four transmembrane helices of the VSD in purple (S1–S4) and the phosphatase domain in peach. **(B)** Reactions catalyzed by VSP. Four different reactions with three different substrates. **(C)** Cartoon representation of a possible VSP dimer organization based on previous results showing both the VSD and phosphatase domains involved in dimerization. **(D)** PDB ID 4G80 structure of Ci-VSD, biological unit, rendered using ChimeraX (Pettersen et al., 2021). The structure suggests a specific dimer interface. Ribbon representation of S1–S4 with S1s in orange and S4s in green. Position F127, I131, I134, and L137 shown in space-filling cartoon. **(E)** Alignment of the VSP S1 helices from several species. Arrows indicate the positions to be mutated to alanine. Sequences for alignments: *laevis*VSP2, gi| 325976472|gb|ADZ48071.1; *laevis*VSP1, gi|148230800|ref|NP_001090072.1; *tropicalis*VSP, gi|62859843|ref|NP_001015951.1; macaqueVSP, gi|75076430|sp| Q4R6N0.1; chickenVSP, gi|118084924|ref|XP_417079.2; mouseVSP, gi|76827498|gb|AAI07330.1; ratVSP, gi|157820295|ref|NP_001406113.1; humanVSP2, gi| 37788781|gb|AAP45144.1; humanVSP1, gi|213972591|ref|NP_954863.2; zebrafishVSP, gi|70887553|ref|NP_001020629.1; cionaVSP, gi|76253898|ref| NP_001028998.1.

(PI(3,4,5)P$_3$) to generate phosphatidylinositol-3,4-trisphosphate (PI(3,4)P$_2$) and from phosphatidylinositol-4,5-trisphosphate (PI(4,5)P$_2$) to generate phosphatidylinositol-4-trisphosphate (PI(4)P). In addition, VSP will remove the 3-phosphate from PI(3,4,5)P$_3$ to generate PI(4,5)P$_2$ and from PI(3,4)P$_2$ to generate PI(4)P. Thus, while VSP catalyzes two types of reactions (3- and 5-phosphate removal), it has three different possible substrates. As a result of being the only enzyme directly coupled to a VSD, VSP serves as a unique member of the family for a better understanding of how the VSD's change conformation in response to changes in the membrane potential.

Because of the importance of S4, many studies have been dedicated to understanding how S4 moves in response to voltage and how it transfers the activation information to the rest of the protein (Freites and Tobias, 2015; Islas, 2016). The S2 and S3 helices are known to contain counter-charges for the S4 argi- nines (Seoh et al., 1996; Papazian et al., 1995). The S1 helix is also known to contribute to the hydrophobic plug (gating pore) that prevents ion conduction through the VSD while still allowing the Arg to move in response to voltage (Chanda and Bezanilla, 2008). However, S1–S3 is mainly considered a scaffold when investigating voltage dependence (Islas, 2016). In contrast, S1 appears to play a more active role in both the voltage-gated proton channel (Hv) and the VSP. Specifically, S1 may help mediate dimerization in Hv (Hong et al., 2015; Li et al., 2015; Lee et al., 2008) and is proposed to play a role in gating Hv's proton

conduction because it moves during the opening transition as opposed to S4, which moves before the opening transition (Mony et al., 2015). In VSP, S1 may also help mediate dimer- ization (Li et al., 2014; Rayaprolu et al., 2018). Importantly, others have shown a functional impact of S1 where amino acid 126 in the *Ciona intestinalis* VSP (Ci-VSP) S1 tunes voltage sensing over two orders of magnitude (Lacroix and Bezanilla, 2012). Mutations in S1 also shift enzyme activity in Ci-VSP (Halaszovich et al., 2012) and in *Danio rerio* VSP (Dr-VSP) (Sakata and Okamura, 2014). In addition, the S1 in Ci-VSP appears to move in response to voltage sensing when the phosphatase is removed (Tsutsui et al., 2013). A complete picture of how S1 influences enzyme activation and voltage sensing in VSP remains unclear. Here, we investigate how VSP uses S1 to control and modulate enzyme activity and the VSD.

The discovery of VSP also boosted our understanding of the modularity of the VSD and has led to the generation of new tools for monitoring and manipulating basic cellular processes. For example, genetically encoded voltage sensor (GEVI) designers immediately took advantage of the VSD from Ci-VSP by con- necting a fluorescent protein directly to the Ci-VSD (Lundby et al., 2008; Baker et al., 2008; Villalba-Galea et al., 2009b). These new generation GEVIs have significantly improved their signal-to-noise ratio allowing researchers to optically monitor single action potentials (Barnett et al., 2012; Piao et al., 2015). Interestingly, the dimerization of fluorescent proteins is known

to impact how GEVIs respond to voltage and Ci-VSD dimerization has been used to design intermolecular Förster resonance energy transfer (FRET) GEVIs (Kang and Baker, 2016). The Ci-VSD has also been attached to other proteins like the "phosphatase and tensin homolog deleted on chromosome 10" (PTEN) (Lacroix et al., 2011). While VSP is a PTEN homolog, PTEN is not voltage-dependent, so the ability to confer voltage dependence onto PTEN indicates the ability to transfer voltage regulation to create new tools for manipulating intracellular reactions. Along the same lines, groups are engineering other VSPs like Dr-VSP to generate a tool for precisely manipulating PIP concentrations in the cell (Kawanabe et al., 2020; Paixao et al., 2023). Understanding how S1 modulates the VSDs will improve the rational design of future VSD-based biosensor tools.

Our current study set out to investigate the contribution of S1 to VSP function. We found that individual mutations in S1 altered the enzyme activity by altering the amount of catalysis, kinetics, and voltage dependence. The same mutations also changed the kinetics and voltage dependence of the S4 motions as measured by voltage clamp fluorometry. When four S1 mutations were combined, further alterations were observed with the enzymatic activity shifted to higher voltages while the S4 motions were shifted to lower voltages, suggesting a decoupling between the VSD and enzyme. When introduced into a GEVI that utilizes the Ci-VSD, the voltage sensitivity and kinetics of the fluorescent signal were also shifted. Thus, S1 plays a strong role in VSP and VSD functions.

## Materials and methods

### Molecular biology
The Ci-VSP in pSD64TF vector was kindly provided by Y. Okamura (Osaka University, Osaka, Japan). fPLC and fTAPP were both kindly provided by E.Y. Isacoff (University of California, Berkeley, Berkeley, CA, USA). The GEVI CC1 was developed by the Baker laboratory (Korea Institute of Science and Technology) (Piao et al., 2015). All mutations and N-terminal epitope tags were added using Pfu Turbo polymerase (Agilent). Primers are listed in Table S1. All DNA was confirmed by sequencing from start to stop codon. cRNA was transcribed using SP6 or T7 mMessage mMachine (Ambion) kits.

### Defolliculating Xenopus laevis oocytes
X. laevis ovaries were purchased from Xenopus One. Briefly, each ovary was washed and then morselized in $Ca^{2+}$-free buffer (96 mM NaCl, 2 mM KCl, 1 mM $MgCl_2$, and 10 mM HEPES, pH 7.6). The morselized ovary was washed in $Ca^{2+}$-free buffer to remove the yolk from lysed oocytes and then digested at room temperature with gentle shaking for 60–70 min in 0.5 mg/ml collagenase type 2 (catalog # LS004177; Worthington Biochemical Corporation) and 0.83 mg/ml collagenase type 3 (catalog # LS004183; Worthington Biochemical Corporation) made in $Ca^{2+}$-free buffer. After digestion, the oocytes were washed in $Ca^{2+}$-free buffer at least 10 times to remove follicles. The oocytes were then sorted and cultured in ND-96 (96 mM NaCl, 2 mM KCl, 1.8 mM $CaCl_2$, 1 mM $MgCl_2$, 50 μg/ml gentamicin, 2.5 mM sodium pyruvate, and 10 mM HEPES, pH 7.6) at 18°C.

### Voltage clamp fluorometry
Voltage clamp fluorometry (VCF) was performed as described previously (Castle et al., 2015). Briefly, X. laevis oocytes were injected with 50 nl of 0.4–0.8 μg/μl of cRNA and incubated in ND-96 at 18°C for 24–36 h. The cRNA dilutions were checked by Nanodrop to ensure the concentrations of total cRNA were correct. On the day of the experiment, oocytes were labeled with tetramethylrhodamine-6-maleimide (TMRM; Invitrogen or Abcam) at a final concentration of 25 μM and the cells were left in the dark on ice for 20 min. Labeled oocytes were then extensively washed and stored in ND-96(–) (ND-96 without gentamicin or pyruvate) at 18–19°C until the end of the experiment. A Leica DM IRBE inverted microscope with a Leica HC Pl APO 20×/0.7 fluorescence objective was used with a Dagan CA-1B amplifier (Dagan Corporation) and illuminated with a Lumen Dynamics X-Cite XLED1 light source. Fluorescence was measured with a ThorLabs photomultiplier tube (PMT). The amplifier and LED were controlled by a Digidata-1440A board and pClamp10.3 software package (Axon Instruments). Light was filtered through an HQ531/40 excitation filter, an HQ593/40 emission filter, and a Q562LP dichroic (Semrock). Fluorescence signals were low-pass filtered at 500 Hz through an eight-pole Bessel filter (Frequency Devices). Glass capillary tubes (VWR) were pulled by a gravity puller (Hamamatsu) for pipette resistances of 0.2–1 MΩ. The VCF voltage protocol consisted of 10 mV steps starting at –150 mV and ending at 200 mV with a holding potential of –80 mV. The measured fluorescence was then plotted against the applied voltage to generate the fluorescence/voltage relationship. All microscopes and optics for electrophysiology and fluorescence measurements were placed on a vibration isolation platform (Newport and TMC) to avoid any vibrational noise and homemade Faraday cages were used to avoid electrical noise.

### Two-electrode voltage clamp electrophysiology and optical measurement of enzymatic activity
Two-electrode voltage clamp (TEVC) was performed as previously described (Castle et al., 2015). FRET-based PIP sensors (Grimm and Isacoff, 2016; Ratzan et al., 2019) were used to measure the depletion of $PI(4,5)P_2$ or the production and depletion of $PI(3,4)P_2$. Specifically, we used the PH domain from phospholipase C (PLC) for $PI(4,5)P_2$ and the PH domain from the tandem PH domain containing protein 1 (TAPP) for $PI(3,4)P_2$. The FRET sensors were designed with an N-terminal CFP and a C-terminal YFP and were called fPLC and fTAPP, respectively. To measure the production of $PI(4,5)P_2$, the diffusion-based sensor GFP-PLC-PH domain (gPLC) was used which has a GFP attached to the N-terminus of the PLC-PH domain. X. laevis oocytes were injected with cRNA mixtures of VSP and biosensor (0.4/0.4 μg/μl for VSP/fPLC; 0.2/0.1 μg/μl for VSP/gPLC; and 0.2/0.4 μg/μl for VSP/fTAPP). The cRNA mixtures were checked by Nanodrop to ensure the concentrations of total cRNA were correct. All VSP constructs for the enzyme activity assays have a TMRM labeling site (G214C) and were tested with VCF to confirm expression. They also have a His-tag to compare with coimmunoprecipitation data. The catalytically inactive Ci-VSP C363S (CS) mutation abolishes catalytic activity and serves as a control for the endogenous X. laevis VSP present in oocytes.

Each cell was injected with 50 nl cRNA mixtures and then incubated in ND-96 for 24–48 h at 18–21°C.

On the day of the experiments, cells were labeled with TMRM as listed under VCF. After labeling, oocytes were stored in ND-96(−) containing 8 µM insulin to promote PI3 kinase activity and upregulate PI(3,4,5)P$_3$ levels. To control for variations in Ci-VSP expression, we tested each cell for VSD motions via VCF as described above. If no VCF was observed, the cell did not move forward for activity testing. For the FRET experiments, light was filtered through a HQ436/20 excitation filter and directed to the objective with a 455LP dichroic (Chroma). The microscope cube did not contain an emission filter because the ThorLabs PMT module contains its own cube with a 510-nm dichroic, an HQ480/40 emission filter for CFP, and an HQ535/30 emission filter for YFP (Chroma). For the gPLC experiments, light was filtered through an HQ470/40 excitation filter, an HQ525/50 emission filter, and a Q496LP dichroic (Chroma). The voltage protocol consisted of steps from −100 to +180 mV in irregular increments. Rest periods of 1–5 min between each voltage step allowed the cell to recover depleted PIP concentrations before the next voltage step. The longer rest periods were for the higher voltages since they were expected to deplete more. Cells remained clamped at the holding potential during all rest periods. The length of the voltage pulse was varied for each biosensor, 2–10 s for fTAPP, 2–20 s for fPLC, and 1.5–3 s for gPLC. The resulting fluorescence was then plotted versus voltage to generate the FV relationship.

### Sensing currents

Defolliculated *X. laevis* oocytes were injected with 50 nl of 0.8 µg/µl (checked by Nanodrop) cRNA of each Ci-VSP S1 mutation, and with the TMRM labeling site mutation (G214C) and a His-tag. These cells were then incubated in ND-96 at 18°C for 24–36 h. Oocytes were labeled with 25 µM TMRM as described in the VCF methods. After labeling, oocytes were held and tested in ND-96(−) + 8 µM insulin as in the enzymatic activity measurements. The VCF setup used was as described in the VCF methods, and all recordings were done at room temperature (22–24°C). The sensing currents were elicited by a 300 ms depolarizing pulse to +150 mV from a holding of −80 mV. The off-sensing current was integrated from the end of the pulse for 150 ms after the pulse. This maximum charge was plotted for each mutation and WT. Leak subtraction was performed by a P/−8 protocol. The simultaneously collected fluorescence was calculated as ΔF/F = [(Fx − F0)/F0]. The absolute value of both the charge and fluorescence data was plotted. Each construct was tested in at least 10 oocytes from at least three biological variable batches of frogs.

### Antibodies and antibody beads

6x-His epitope tag antibody (His.H8) was purchased from Invitrogen. Monoclonal Flag M2 antibody (F1804-50UG) and anti-Flag M2 affinity gel (A2220-1ML) were purchased from Sigma-Aldrich. Goat anti-mouse light chain–only HRP was purchased from Jackson Labs (115-035-174).

### Co-immunoprecipitation (co-IP)

Oocytes were injected with 50 nl of 0.4–0.8 µg/µl of total cRNA for each construct. Individual constructs were at 0.4 µg/µl and

mixtures at 0.4/0.4 µg/µl. The cRNA dilutions and mixtures were checked by Nanodrop to ensure the concentrations of total cRNA were correct. Oocytes were incubated for 24–36 h in ND-96 at 18–21°C. Protein expression was confirmed using VCF. Post-expression check, 1–2 oocytes of those injected with a single RNA and 2–3 oocytes of those injected with two RNAs were lysed in 200 µl of lysis buffer (150 mM NaCl, 0.1% IGEPAL, 50 mM Tris-base, pH 7.4 with protease inhibitors). The lysates were centrifuged at 21,100 *g* for 15 min at 4°C. The supernatant (160 µl) was split into two parts (one aliquot of 130 µl and one aliquot of 30 µl). The anti-FLAG beads in lysis buffer (100 µl, 1:20 ratio) were added to the 130 µl sample aliquot. These mixtures were left on a rotator overnight at 4°C. The 30 µl aliquot was stored at −20°C and used as an input control. The samples with the beads were then centrifuged at 1,200 *g* for 5 min at 4°C to pellet the beads. The supernatant was discarded and the beads were washed with lysis buffer five times. After the last wash, 7 µl of 3X Laemmli sample buffer was added directly to the beads. To the input control, 7 µl of 5X Laemmli sample buffer was added and all samples were incubated for 5 min at 98°C. The samples were loaded on a 10% SDS polyacrylamide gel (10 µl/lane) and electrophoresed at 200 mV for about 45 min until the bromophenol blue in the sample buffer reached the bottom of the gel.

### Western blotting

The samples were transferred from the polyacrylamide gel onto a nitrocellulose membrane at 350 mA for 70 min using a sodium borate buffer system. After transfer, the membranes were incubated in a blocking buffer (5% milk in 1X TBS-T) for 30 min. The membranes were then incubated with the primary antibody solutions (1:2,000 dilution prepared in blocking buffer) for 1 h at room temperature. Membranes were washed six to eight times with 1X TBS-T over a period of about 1 h. Secondary antibody solutions prepared in blocking buffer (1:15,000 dilution) were added to the membranes and incubated for 1 h at room temperature. Washes with 1X TBS-T were repeated six to eight times. Membranes were developed using WesternBright ECL reagent (Advansta).

### Patch clamp electrophysiology: Voltage imaging with GEVIs

The original CC1 construct with super ecliptic pHluorin (SE) A227D (Piao et al., 2015) or CC1 variants with mutations in S1 were expressed in HEK 293 cells via lipofectamine (Invitrogen). Coverslips with transiently transfected cells were placed into a bath chamber (Warner Instruments) kept at 34°C throughout the experiment and perfused with bath solution (150 mM NaCl, 4 mM KCl, 1 mM MgCl$_2$, 2 mM CaCl$_2$, 5 mM D-glucose, and 5 mM HEPES, pH = 7.4). Filamented glass capillary tubes (1.5/0.84 mm; World Precision Instruments) were pulled by a micropipette puller (Sutter) prior to each experiment with pipette resistances of 3–5 MΩ. The pipettes were filled with intracellular solution (120 mM K-aspartate, 4 mM NaCl, 4 mM MgCl$_2$, 1 mM CaCl$_2$, 10 mM EGTA, 3 mM Na$_2$ATP, and 5 mM HEPES, pH = 7.2). Whole-cell voltage clamp was conducted using a patch clamp amplifier (HEKA). A holding potential of −70 mV was used for all recordings.

An inverted microscope (IX71; Olympus) equipped with a 60× oil-immersion lens with 1.35-numerical aperture was used for

epifluorescence imaging. The light source was a 75 W Xenon arc lamp (Osram) placed in a lamp housing (Cairn). GFP was imaged using a filter cube consisting of an excitation filter (FF02-472/30-25), a dichroic mirror (FF495-Di03), and an emission filter (FF01-496) for the 470 nm wavelength excitation (Semrock). Fluorescence changes of the voltage indicators were typically recorded at a 1 kHz frame rate by a high-speed CCD camera (RedShirtImaging). All the relevant optical devices were placed on a vibration isolation platform (Kinetic Systems) to avoid any vibrational noise during patch clamp fluorometry experiments.

### Data analysis

ChimeraX was used to visualize the structure of the Ci-VSD (Pettersen et al., 2021; Li et al., 2014). Axon Clampfit 10.3 (Molecular Devices) was used to initially process the fluorescence and FRET data. For the VCF, the fluorescence baseline was zeroed and then the amplitude of each pulse was measured. The voltages for each pulse were measured. The expression VCF data was calculated as $\Delta F/F = [(Fx – F0)/F0]$. Current data was processed in Clampfit. Statics were performed in Microsoft Excel with the Student's test calculated as two-tailed and used for equal variance data while the Welsh's $t$ test was calculated as two-tailed and used for unequal variance data. The FRET ratio was also calculated in Clampfit. Subsequent analysis of the TEVC activity data was automated using R scripts in R Studio 4.0.2 and edited in 4.1.1 (deposited in GitHub, https://github.com/jtkelley46/VSP). Specifically, for the FRET or fluorescence increase (fTAPP and gPLC, respectively), the activity $\Delta F/F$ was calculated from the prepulse baseline to the max signal increase. For the fPLC FRET decrease, the activity $\Delta F/F$ was calculated from the prepulse baseline to the max FRET decrease. For the fTAPP FRET decrease, the activity $\Delta F/F$ was calculated from the max FRET increase to the max FRET decrease. In some instances, oocytes injected with catalytically inactive Ci-VSP C363S showed significant background activity indicating that the native *X. laevis* VSP was contributing to the activity signal (Ratzan et al., 2011; Ratzan et al., 2019). For those traces, we subtracted an average trace from oocytes expressing C363S from the traces for oocytes expressing wild type (WT) and the mutant Ci-VSP. All final data were plotted in Igor Pro. Final voltage-dependent curves were plotted with change in FRET or fluorescence on the Y-axis and measured voltage on the X-axis. Igor Pro was used to calculate all fits: single or double Boltzmann sigmoid, single or double exponentials, and residuals. Activity assays were repeated on oocytes extracted on different days from at least three different frogs (biological replicates) until data from a minimum of eight oocytes were acquired and analyzed. Error bars indicate the standard error of the mean (SEM). Figures were made in Illustrator (Adobe).

For the patch fluorometry experiments, images were acquired by NeuroPlex (RedShirtImaging) and analyzed for initial parameters such as fluorescence change ($\Delta F = Fx – F0$) or fractional fluorescence change values ($\Delta F/F = [(Fx – F0)/F0] * 100$) by Microsoft Excel. The acquired data from whole-cell voltage clamp experiments of HEK 293 cells were averaged for 16 trials. Data were collected from recorded cells that did not lose their seals during the whole-cell voltage clamp recording. $\Delta F/F$ values for the tested voltage pulses were plotted using Igor Pro.

### Online supplemental material

Fig. S1 includes raw kinetic activity data for fTAPP and fPLC sensors with VSP mutations before subtraction. It also includes activity versus voltage relationships for the short pulse S1-Q activity using the fTAPP biosensor as well as the comparison of the inactive VSP versus endogenous VSP activity. Fig. S2 includes poltos of the amplitudes from VCF expression tests for each mutation tested. It also includes a comparison of VCF amplitudes with the sensing current amplitudes from the same cells. Fig. S3 includes the kinetic analysis of the VCF data for WT, F127A, L137A, and S1-Q mutations versus voltage. It also includes the single versus double Boltzmann sigmoid comparison in the form of residuals. Table S1 lists the primers used to mutate the VSP DNA. Table S2 lists the τ values for the kinetic analysis of the VCF data for all the mutations.

## Results

To determine the role the S1 helix plays in the Ci-VSP function, we started with the Ci-VSD crystal structure that was solved in a dimer conformation (Li et al., 2014). Specifically, the biological unit of the structure showed two adjacent VSDs with the S1 and S4 helices of one subunit facing the S1 and S4 helices of the other subunit. This conformation is consistent with a subsequent study from our lab which found that full-length Ci-VSP is able to dimerize at high concentrations (Rayaprolu et al., 2018). The same study showed that both the VSD and the phosphatase domain can dimerize independently, suggesting the overall organization of the dimer could be a side-by-side orientation with both domains involved in dimerizing the full-length VSP (Fig. 1 C). We chose to manipulate sites on S1 that lined the interface between the two subunits, F127, I131, I134, and L137, for two reasons (Fig. 1 D). First, we expect dimers to influence Ci-VSP activity as seen previously (Rayaprolu et al., 2018), so we wanted to test positions that may be involved in transmitting information across the dimer. Second, we wanted to minimize directly altering the S4 contact points between S1 and S4 (either intra or intersubunit). Of the selected sites, both the F127 and I134 positions exhibited high conservation of hydrophobic residues across different species homologs of VSPs (Fig. 1 E). L137 is nearly invariable across different VSPs, while I131 is more tolerant of hydrophobic substitutions (I, F, and V) and hydrophilic (T) residues found in the equivalent positions.

First, we tested the impact of the individual mutations on the enzymatic activity. Pleckstrin homology (PH) domains from the tandem PH domain-containing protein 1 (TAPP) and from PLC selectively bind $PI(3,4)P_2$ and $PI(4,5)P_2$, respectively, making them ideal sensors for Ci-VSP activity. We used two different biosensor designs with these PH domains, one where the signal is dependent on a FRET signal and one where the signal is dependent on the free diffusion of the PH domain (diffusion-based). The TAPP FRET sensor, called fTAPP, has an N-terminal CFP followed by the TAPP-PH domain and then a C-terminal YFP with a prenylation site for membrane localization (Fig. 2 A) (Grimm and Isacoff, 2016; Sato et al., 2003). An increase in FRET follows the 5-phosphatase reaction of $PI(3,4,5)P_3$ to $PI(3,4)P_2$ while a decrease in FRET indicates the 3-phosphatase

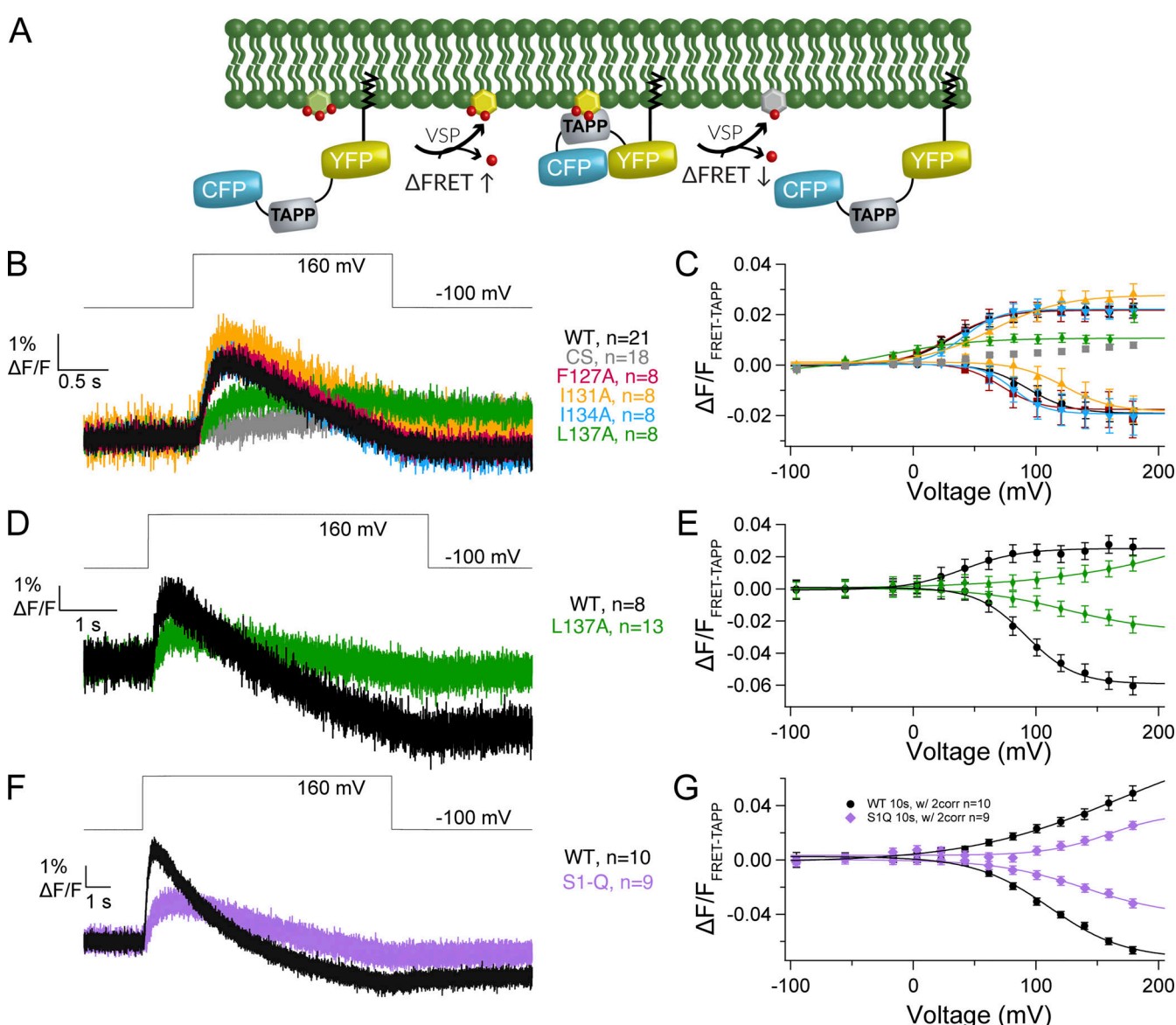

Figure 2. **S1 helix modulates kinetics and voltage dependence of PI(3,4,5)P₃ and PI(3,4)P₂ dephosphorylation. (A)** Cartoon depicting the biosensor fTAPP's expected response to VSP 5-phosphate dephosphorylation from PI(3,4,5)P₃ (increase in FRET) and 3-phosphate dephosphorylation from PI(3,4)P₂ (decrease in FRET). **(B)** Averaged fTAPP data for a voltage step from −100 mV holding the potential to 160 mV for 2 s for the single S1 mutations (F127A, I131A, I134A, and L137A). L137A appears to only show 5-phosphatase activity while the others closely resemble WT kinetics of activity. **(C)** Voltage-dependent fTAPP activity for the single S1 mutations. The up-FRET component was separated from the down-FRET component (see Materials and methods). While F127A and I134A are very similar to WT, the I131A mutation shifts the voltage dependence of both reactions to higher voltages and the L137A mutation only displays 5-phosphatase activity with the short 2-s steps. **(D)** Averaged data for a longer voltage step (5 s) from −100 to 160 mV for L137A and WT with the CS control subtracted from each (data without subtraction in Fig. S1 B). A clear FRET decrease is visible for L137A at the longer step, indicating that while the mutation does not eliminate 3-phosphatase activity, it slows it down relative to WT. **(E)** L137A voltage-dependent fTAPP activity for the 5-s voltage step. Both up and down FRET components are visible though the upward component appears more linear. **(F)** Averaged data for a longer voltage step (10 s) from −100 to 160 mV for S1-Q and WT with the CS control subtracted from each (data without subtraction in Fig. S1 E). A clear FRET decrease is visible for S1-Q at the longer step, indicating that while the mutation does not eliminate 3-phosphatase activity, it slows it down relative to WT. **(G)** S1-Q and WT voltage-dependent fTAPP activity for the 10-s voltage step. Both up and down FRET components are visible though the upward component appears more linear. All error bars are ± SEM. Data fit with single Boltzmann sigmoid equations.

reaction of PI(3,4)P₂ to PI(4)P (Fig. 2 A). To increase the amount of PI(3,4,5)P₃ in the cells, all enzymatic activity is recorded in a solution of insulin to activate the endogenous insulin receptor and the subsequent phosphatidyl-3-kinase. WT Ci-VSP gives a robust increase and decrease in FRET when tested with fTAPP consistent with previous publications (Fig. 2 B) (Rayaprolu et al.,

2018; Grimm and Isacoff, 2016; Paixao et al., 2023). Because *X. laevis* oocytes are known to express endogenous VSPs (Xl-VSP1 and 2) (Ratzan et al., 2019; Castle et al., 2015; Kohout et al., 2010), a negative control is also needed. Previously, we used an inactive Ci-VSP using the catalytic site mutation, C363S (CS), to mimic our experimental conditions (Castle et al., 2015; Rayaprolu et al.,

Table 1.  fTAPP biosensor activity

| | $n$ | Pulse time (s) | $PI(3,4,5)P_3$ to $PI(3,4)P_2$ | | $PI(3,4)P_2$ to $PI(4)P$ | |
|---|---|---|---|---|---|---|
| | | | $V_{1/2}$ up | Up slope | $V_{1/2}$ down | Down slope |
| WT | 21 | 2 | 32 ± 2 | 20 ± 2 | 94 ± 4 | 17 ± 3 |
| F127A | 8 | 2 | 30 ± 2 | 20 ± 2 | 72 ± 3 | 15 ± 2 |
| I131A | 9 | 2 | 65 ± 3 | 29 ± 3 | 124 ± 6 | 19 ± 4 |
| I134A | 8 | 2 | 41 ± 2 | 15 ± 2 | 82 ± 2 | 13 ± 2 |
| L137A | 8 | 2 | 21 ± 10 | 34 ± 7 | ns | ns |
| S1-Q | 8 | 2 | 222 ± 554 | 193 ± 193 | ns | ns |
| WT | 9 | 5 | 41 ± 4 | 23 ± 4 | 92 ± 2 | 20 ± 2 |
| L137A | 13 | 5 | >276 | 71 ± 45 | 124 ± 9 | 36 ± 6 |
| WT | 10 | 10 | 167 ± 17 | 62 ± 6 | 111 ± 4 | 30 ± 3 |
| S1-Q | 9 | 10 | 159 ± 40 | 25 ± 15 | 140 ± 13 | 36 ± 6 |

ns = no signal detected.

2018; Liu et al., 2012). Others use the fTAPP sensor alone to measure endogenous Xl-VSP (Paixao et al., 2023; Kurokawa et al., 2012). Since a side-by-side comparison had not been done to our knowledge, we tested both CS with fTAPP and fTAPP alone using long, 10-s pulses. We found that the CS with fTAPP resulted in statistically higher Xl-VSP activity compared with fTAPP alone (Fig. S1 A), countering the idea that the two conditions are equivalent. Interestingly, our previous research on the Xl-VSPs showed that Xl-VSP1 can catalyze both $PI(3,4,5)P_3$ and $PI(3,4)P_2$ as substrates while Xl-VSP2 is only active against $PI(3,4,5)P_3$ as a substrate. These results suggest that Xl-VSP2 may dominate the endogenous VSP signal since we only see activity against $PI(3,4,5)P_3$. Because of the observed difference when CS was expressed, we coexpressed the sensors with CS to control for our experimental conditions more accurately. As seen before, we observe only a small amount of signal coming from endogenous Xl-VSPs when we use 2-s depolarizations since the kinetics of the Xl-VSP2 are very slow (Fig. 2, B and C) (Rayaprolu et al., 2018; Ratzan et al., 2019).

When testing the S1 mutations, the kinetics for the F127A and I134A mutations were very similar to those of WT (Fig. 2 B). The voltage dependence for F127A was similar for the 5-phosphatase reaction while slightly shifted to lower voltages for the 3-phosphatase activity (Fig. 2 C and Table 1). I134A showed a higher voltage requirement for the 5-phosphatase reaction and a slightly lower voltage requirement for the 3-phosphatase reaction. Interestingly, I131A, which is the most variable position (Fig. 1 E), also showed similar kinetics but with a shift to more positive potentials for both reactions, indicating this position's variability could be part of tuning the voltage dependence between species. The highly conserved L137 position meanwhile dramatically changed activity when mutated to an alanine. The 5-phosphatase activity was greatly reduced and the 3-phosphatase activity was not visible using a 2-s voltage step (Fig. 2, B and C). However, when the pulse was extended out to 5 s, a small 3-phosphatase activity was seen, indicating that the L137A did not eliminate the 3-phosphatase activity but significantly slowed

down the reaction compared to the 5-phosphatase reaction (Fig. S1 B). The longer pulses also resulted in significantly higher endogenous VSP activity as measured by the inactive CS mutation (Fig. S1 B), obscuring the true activity of the L137A. To compensate for the endogenous activity, we subtracted the CS data from the L137A and WT data, as was done previously (Liu et al., 2012). The resulting traces showed a robust though slower decrease in FRET when compared with WT (Fig. 2 D), indicating both reactions were still present. Interestingly, both reactions were significantly shifted to higher voltages and are reduced in magnitude compared with WT (Fig. 2 E). The 5-phosphatase reaction appears to approach a more linear relationship with voltage, with its larger slope suggesting a weaker sensitivity to voltage and potentially weaker coupling between the VSD and the enzyme or between dimers. The 3-phosphatase reaction retains a clearer sigmoidal relationship with voltage. The strong impact of mutating L137 is consistent with it being well conserved across species and indicates that leucine in position 137 plays a significant role in modulating VSP function (Fig. 1 E).

When all four mutations were combined in S1-Q and tested at 2-s pulses, the 3-phosphatase activity was not visible, similar to the L137A data (Fig. S1 C). The 5-phosphatase activity was greatly reduced though still above the CS negative control (Fig. S1 D). Longer 10-s pulses revealed 3-phosphatase activity (Fig. S1 E), with the endogenous Xl-VSPs giving a large signal that obscures the impact of S1-Q. As with the L137A alone data, we subtracted the CS background from the WT and S1-Q longer pulses. The resulting traces show both reactions clearly though with much slower kinetics (Fig. 2 F). Interestingly, while the 5-s pulses did not dramatically change the WT response, the 10-s pulses shifted the WT voltage dependence to higher voltages for the 5-phosphatase reaction and the 3-phosphatase reaction remained similar (Fig. 2 G and Table 1). For the S1-Q mutation, its 5-phosphatase reaction gives a similar voltage dependence to WT though the amplitude is significantly reduced. The S1-Q 3-phosphatase reaction is shifted to higher potentials as well (Table 1). These data show the importance of helices other than S4 to modulate the ability of the

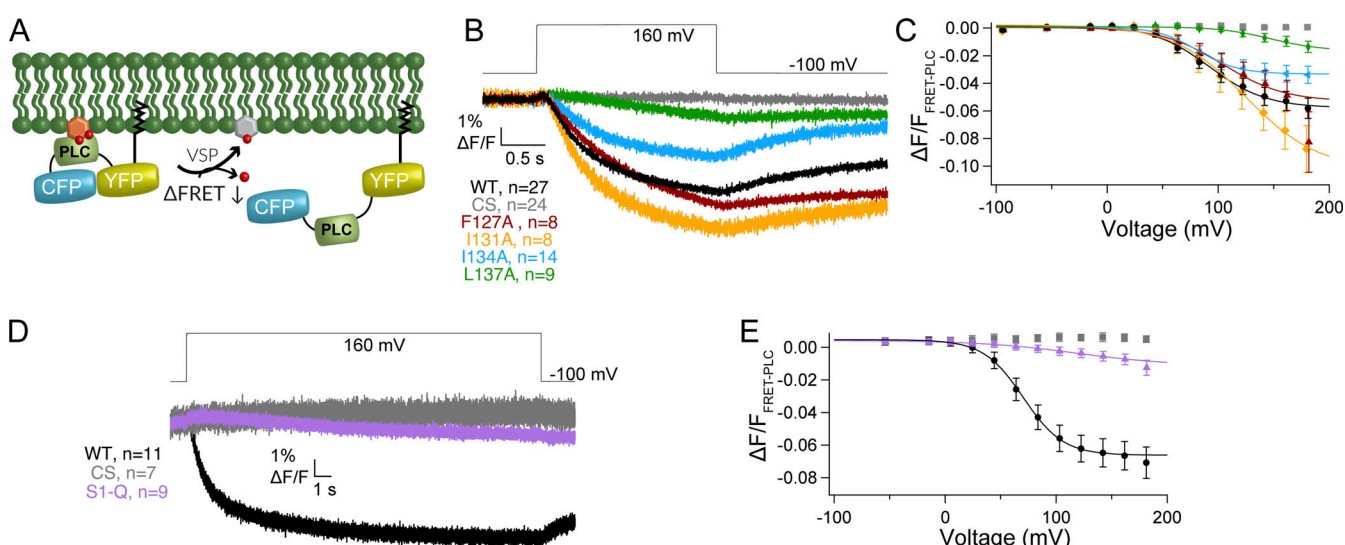

**Figure 3.** **S1 helix impacts the kinetics of PI(4,5)P$_2$ 5-phosphate dephosphorylation. (A)** Cartoon depicting the biosensor fPLC's expected response to VSP 5-phosphate dephosphorylation from PI(4,5)P$_2$ (decrease in FRET). **(B)** Averaged fPLC data for a 2-s voltage step from −100 to 160 mV for the single S1 mutations (F127A, I131A, I134A, and L137A). L137A significantly slowed the 5-phosphatase activity while the other mutations more closely resemble WT kinetics. **(C)** Voltage-dependent fPLC activity for the single S1 mutations. The I131A mutation shifts the voltage dependence of activity to higher voltages. The L137A mutation displays reduced and right-shifted 5-phosphatase activity. **(D)** Averaged data for a long voltage step (20 s) from −100 to 160 mV for S1-Q and controls. A FRET decrease is visible for S1-Q at the longer step, indicating that while the mutation does not eliminate 5-phosphatase activity, it significantly slows it down. The CS constructs indicates minimal endogenous Xl-VSP2 activity even with such a long pulse. **(E)** Voltage-dependent fPLC activity for S1-Q, WT, and CS using the longer (20-s) pulses. While S1-Q is still active, the activity is almost linear with voltage dependence. All error bars are ± SEM. Data fit with single Boltzmann sigmoid equations.

phosphatase to catalyze reactions. It also demonstrates the importance of different pulse lengths for characterizing VSP function.

The design of the PLC FRET-based biosensor, fPLC, is similar to that of fTAPP but the PH domain from PLC is used instead since PLC-PH is selective for PI(4,5)P$_2$ (Fig. 3 A). We did not use fPLC to quantify the production of PI(4,5)P$_2$ from PI(3,4,5)P$_3$ because, in our hands (and others), the signal is too low for reliable quantification in individual traces (Rayaprolu et al., 2018; Paixao et al., 2023; Ratzan et al., 2019). However, fPLC is still a good and fast biosensor for quantitating the 5-phosphatase reaction, from PI(4,5)P$_2$ to PI(4)P, via a decrease in the FRET signal (Fig. 3 A). Thus, using fPLC, we observed the expected FRET decrease with WT and little to no FRET decrease with the CS negative control (Fig. 3 B). We found that both I131A and I134A had similar fPLC activity kinetics compared with WT while F127A and L137A were slower. L137A also had lower overall activity and was significantly shifted to more positive voltages (Fig. 3, B and C; and Table 2). F127A and I134A had similar voltage-dependent activities compared with WT while I131A was again shifted to higher voltages. When S1-Q was tested with fPLC using a 2-s pulse, it appeared that the mutation eliminated all activity (Fig. S1 F). When a significantly longer pulse was given (20 s), we did observe a small amount of PI(4,5)P$_2$ to PI(4)P activity above the negative control values though it was slower, reduced, and right-shifted (Fig. 3, D and E; and Table 2). Overall, the fPLC data were similar to the fTAPP data in that the most dramatic impacts are from the L137A alone mutation and the S1-Q combination. Both slow down the kinetics with S1-Q

significantly slower than L137A even and both shift the voltage dependence to much higher potentials.

To monitor the production of PI(4,5)P$_2$ from PI(3,4,5)P$_3$, we used the diffusion-based GFP-PLC-PH biosensor (gPLC), where the increase in fluorescence at the membrane indicates the production of PI(4,5)P$_2$ from PI(3,4,5)P$_3$ (Fig. 4 A) (Castle et al., 2015; Balla 2013). As seen previously, WT gives a robust gPLC fluorescence increase and CS shows no endogenous Xl-VSP signal using either a 1.5-s or a 3-s depolarization (Fig. 4 B). However, we see a different overall pattern for the mutations compared with the previous biosensors. With the S1 mutations, we found that F127A, I131A, and L137A were slower than WT with L137A requiring a longer 3-s pulse to see the full signal (Fig. 4 B). Interestingly, L137A has not lost as much overall activity or voltage dependence as is seen with the other reactions (Fig. 4 C). Both F127A and I131A shifted the voltage dependence of activity to higher voltages, requiring more energy to activate, while L137A slightly shifted the voltage dependence to lower voltages (Fig. 4 C and Table 2). I134A has similar kinetics and voltage activation compared with WT; however, it appears more active than WT with a larger amplitude fluorescence change. When all four mutations were combined in S1-Q, we observed dephosphorylation at the 3-phosphate from PI(3,4,5)P$_3$ again with slower kinetics (Fig. 4 D). The voltage dependence of S1-Q appears significantly shifted to higher voltages and shows a significantly smaller amount of catalysis (Fig. 4 E). These results indicate that the identity of the substrate is important for voltage-dependent catalysis. In addition, the PI(3,4,5)P$_3$ to PI(4,5)P$_2$ reaction was quite different for the individual S1 mutation

**Table 2.  gPLC and fPLC biosensor activity**

| | gPLC | | | | fPLC | | | |
| | PI(3,4,5)P₃ to PI(4,5)P₂ | | | | PI(4,5)P₂ to PI(4)P | | | |
| | *n* | Pulse time (s) | $V_{1/2}$ | Slope | *n* | Pulse time (s) | $V_{1/2}$ | Slope |
|---|---|---|---|---|---|---|---|---|
| WT | 20 | 1.5 | 90 ± 3 | 28 ± 2 | 27 | 2 | 92 ± 3 | 26 ± 3 |
| F127A | 8 | 1.5 | 137 ± 6 | 36 ± 3 | 8 | 2 | 104 ± 5 | 28 ± 4 |
| I131A | 9 | 1.5 | 104 ± 7 | 44 ± 5 | 8 | 2 | 125 ± 6 | 33 ± 4 |
| I134A | 9 | 1.5 | 89 ± 2 | 20 ± 1 | 14 | 2 | 81 ± 2 | 18 ± 2 |
| L137A | 8 | 3 | 64 ± 7 | 50 ± 8 | 9 | 2 | 147 ± 14 | 23 ± 8 |
| S1-Q | 9 | 3 | 167 ± 20 | 58 ± 7 | 11 | 2 | 181 ± 91 | 15 ± 22 |
| WT | N/A | N/A | N/A | N/A | 11 | 20 | 70.2 ± 0.7 | 18.0 ± 0.7 |
| S1-Q | N/A | N/A | N/A | N/A | 9 | 20 | 119 ± 13 | 44 ± 8 |

compared with the other biosensors while the S1-Q catalysis was more like the other S1-Q reactions.

Before each activity assay, every cell was tested for a VCF signal to ensure that the VSP was expressed on the membrane. If no VCF signal was observed, the cell was discarded and another one was tested. All this expression data was compiled and no significant difference was observed for the VCF signal from each mutation when compared with WT (Fig. S2 A). VCF is well established to monitor both the sensing current and subsequent protein conformational changes leading to the relaxed state of the VSD (Villalba-Galea et al., 2008). VCF does have limitations, however. Very small amounts of protein will not give a fluorescence signal and Western blots are needed to establish protein expression (Rayaprolu et al., 2018). In addition, the amount of protein on the membrane is not easily quantitated from the fluorescence change. When Ci-VSP is expressed at very low concentrations, the activities are known to become shifted to lower voltages (Rayaprolu et al., 2018). To ensure our changes in voltage-dependent activity were due to intrinsic properties of the mutation and not as a result of significantly lower protein expression, we also tested the sensing charge movement and VCF signal from the same cells for each mutant (Fig. S2, B and C). We found that three mutations showed less sensing charge than WT, F127A, I134A, and S1-Q, suggesting that these proteins were not as abundant on the membrane compared with WT. However, the VCF signals from all the same cells were similar to WT, except for CS and S1-Q, which gave larger fluorescence values than WT. Thus, the lower expression does not account for the observed shifts in activity.

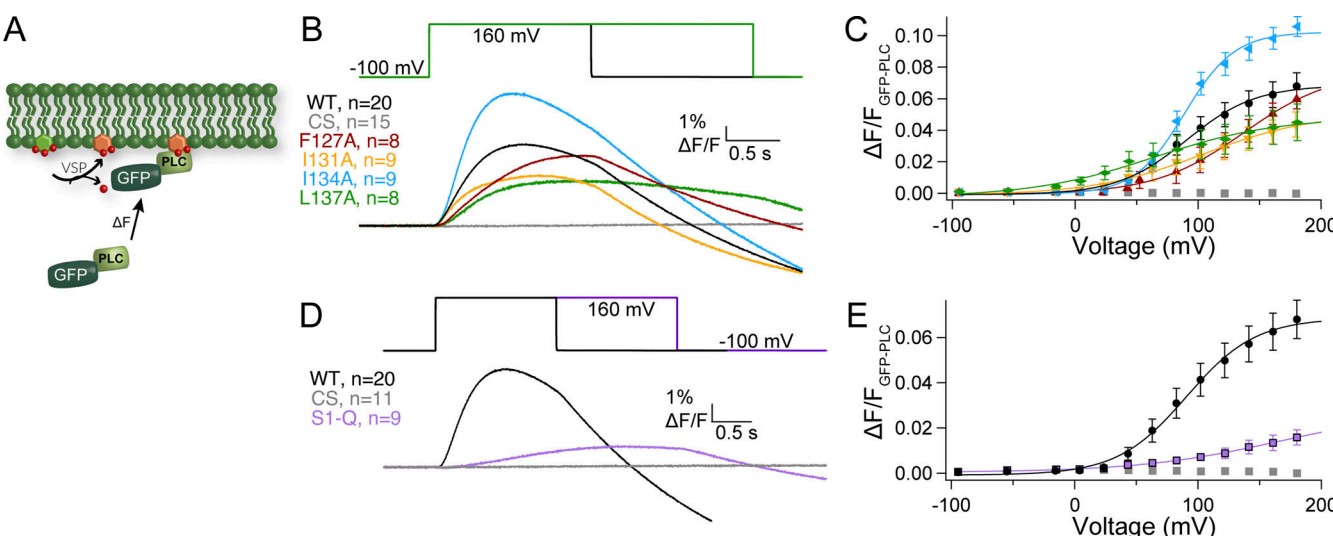

**Figure 4.  S1 helix modulates the kinetics of PI(3,4,5)P₃ 3-phosphate dephosphorylation. (A)** Cartoon depicting the biosensor gPLC's expected response to VSP 3-phosphate dephosphorylation from PI(3,4,5)P₃ to PI(4,5)P₂ (increased membrane fluorescence). **(B)** Averaged gPLC data for a voltage step from −100 to 160 mV for the single S1 mutations (F127A, I131A, I134A, and L137A). L137A significantly slowed the 3-phosphatase activity requiring a 3-s pulse. **(C)** Voltage dependent gPLC activity for the single S1 mutations. While I134A is similar to WT voltage dependence, it shows an enhanced overall activity. Both the F127A and I131A mutations shift the voltage dependence of activity to higher voltages while L137A displays reduced 3-phosphatase activity and is slightly left shifted though the slope is significantly reduced. **(D)** Averaged data for a voltage step from −100 to 160 mV for WT and S1-Q. Slower activation was observed for S1-Q so a 3-s pulse was used. **(E)** Voltage-dependent gPLC activity for WT and S1-Q. The S1-Q significantly reduced the activity and shifted the voltage dependence to higher voltages, more than just the sum of the individual mutations. All error bars are ± SEM. Data fit with single Boltzmann sigmoid equations.

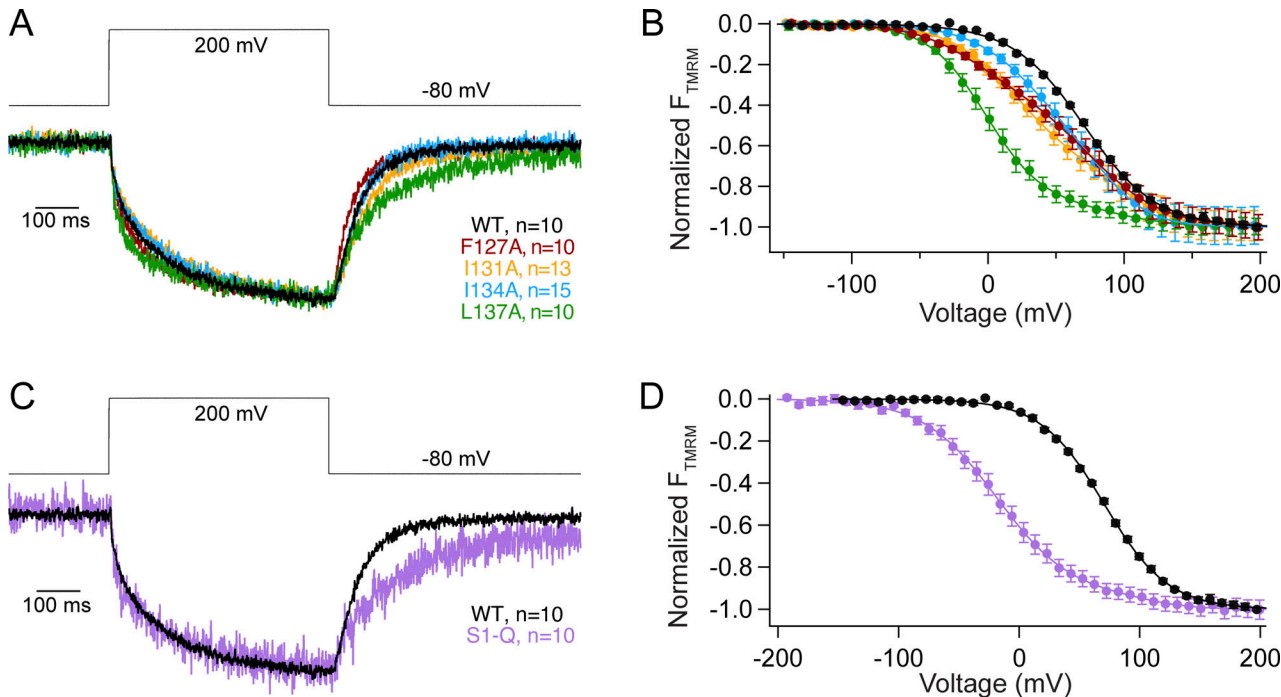

**Figure 5. S1 helix tunes voltage dependence of S4 motion. (A)** Representative VCF traces for a voltage step from −80 to 200 mV for all individual S1 mutations. Activation and repolarization kinetics are similar to WT except for L137A which has a second, slower repolarization motion. **(B)** Voltage dependence of the VSD motions. Mutations in S1 of VSP shift the voltage dependence of VSD motions to lower voltages and introduce a second component in the motion. WT fit by single Boltzmann sigmoid while mutants are fit by double Boltzmann sigmoid. **(C)** Representative VCF trace for a voltage step from −80 to 200 mV for S1-Q and WT. Repolarization kinetics show an additional slower component for S1-Q. **(D)** The voltage dependent motions of S1-Q significantly shifted the voltage dependence of activation by almost 90 mV compared to WT. It also introduced a second component. WT fit by single Boltzmann sigmoid, and S1-Q fit by double Boltzmann sigmoid. All constructs have an N-terminal His-tag to compare with co-IP data. All errors are ± SEM.

Using VCF, we then tested how each individual mutation impacted VSD motions. We used the common labeling site, G214C, in the S3–S4 loop since it has been shown to be a good reporter of both sensing charge movement and VSD motions (Kohout et al., 2008; Villalba-Galea et al., 2008). The activation kinetics for WT and all the mutations were well fit by a double exponential as previously reported (Fig. 5 A and Table S2) (Kohout et al., 2008; Villalba-Galea et al., 2008). We found that the first rate, $\tau_{a1}$, which is correlated with the sensing charges, is similar between WT and all the single mutations (Table S2). However, the second rate, $\tau_{a2}$, which is correlated with the VSD relaxed state, was slightly faster in both F127A and L137A compared with WT (Table S2). We further analyzed the F127A and L137A kinetics across all the tested voltages and found a slightly faster $\tau_{a1}$ across multiple voltages (Fig. S3 A). However, the overall voltage dependence of either rate ($\tau_{a1}$ or $\tau_{a2}$) was not significantly different. The repolarization kinetics for F127A, I131A, and I134A were similar to WT and were well described by single exponential fits, suggesting that overall, they don't significantly influence the kinetics of S4 motions. However, the repolarization kinetics for L137A were slower and better fit by a double exponential, where the first rate, $\tau_{r1}$, is similar to WT and the second rate, $\tau_{r2}$, was significantly slower (Fig. 5 A, Table S2, and Fig. S3 B). Interestingly, $\tau_{r1}$ consistently slowed from 0 to 70 mV for both WT and L137A. Above 70 mV, $\tau_{r1}$ stayed the same, indicating no further voltage dependence. The second repolarization rate, $\tau_{r2}$, for L137A was also not dependent on voltage and

could represent an added step for the VSD to exit the relaxed state. In addition to the kinetic differences, each individual S1 mutation lowered the voltage dependence of the steady state motions, and they were all best fit by a double Boltzmann sigmoid (Fig. 5 B and Table 3). To confirm the double Boltzmann sigmoid was required, we calculated the residuals for each trace and found that the double sigmoid consistently fit the data better than the single sigmoid (Fig. S3 C). These results suggest that S1 directly impacts the kinetic and voltage-dependent motions of S4. Both F127A and L137A shifted the voltage dependence of $V1_{1/2}$ to lower voltages, by 81 and 72 mV, respectively. Interestingly, F127A $V2_{1/2}$ corresponds well with the WT single $V_{1/2}$ while L137A $V2_{1/2}$ shifted to higher voltages by 28 mV. Both I131A and I134A shifted $V1_{1/2}$ to lower voltages by smaller amounts while their $V2_{1/2}$ also shifted to higher voltages (Table 3).

When all four mutations were combined in S1-Q, both activation rates, $\tau_{a1}$ and $\tau_{a2}$, were slightly faster than WT at 200 mV (Table S2), though that trend was not consistent across multiple voltages (Fig. S3 A). For repolarization, the S1-Q $\tau_{r1}$ appeared slower than WT though the difference was not significant for all voltages (Fig. 5 C and Fig. S3 B). Interestingly, the S1-Q $\tau_{r1}$ appeared less dependent on voltage particularly in the lower voltages (0–70 mV) (Fig. S3 B). The S1-Q $\tau_{r2}$ was not voltage dependent and overall is similar to the L137A $\tau_{r1}$. The S1-Q voltage dependence was also a strongly shifted, double Boltzmann sigmoid with $V1_{1/2}$ shifted by 88 mV to lower voltages and $V2_{1/2}$ shifted by 37 mV to higher voltages (Fig. 5 D and Table 3).

**Table 3. VCF, voltage dependent S4 motions**

|  | *n* | V1$_{1/2}$ | Slope 1 | V2$_{1/2}$ | Slope 2 |
|---|---|---|---|---|---|
| WT | 10 | ns | ns | 70.5 ± 0.5 | 26.5 ± 0.4 |
| F127A | 10 | −11 ± 4 | 24 ± 2 | 76 ± 3 | 27 ± 1 |
| I131A | 13 | 20 ± 3 | 25 ± 1 | 100 ± 5 | 20 ± 3 |
| I134A | 15 | 37 ± 5 | 25 ± 1 | 89 ± 3 | 16 ± 2 |
| L137A | 10 | −2 ± 1 | 21.2 ± 0.7 | 99 ± 11 | 23 ± 8 |
| S1-Q | 10 | −18 ± 2 | 31 ± 1 | 108 ± 14 | 20 ± 12 |

ns = no signal detected.

Since the amino acids tested were at the interface of the VSD dimer, we needed to know whether the mutations altered dimerization to be able to interpret their impact on VSP. Using His and FLAG tags to differentiate between the two subunits of the dimers (Fig. 6 A), we pulled down with anti-FLAG beads. They all showed pulldown (Fig. 6 B), indicating that none of the individual mutations altered Ci-VSP dimerization. We only tested the pulldown of the His subunit with the anti-FLAG beads because we discovered that Ci-VSP occasionally binds nonspecifically to cobalt and nickel beads. S1-Q also showed pull down (*n* = 4). We did observe one blot that did not show any pulldown (*n* = 1) even though the input protein concentrations were similar (Fig. 6 C). This variability could suggest that the mutations may slightly weaken the interactions between the Ci-VSP subunits, but they do not eliminate them. Thus, these mutations can serve to explore the contribution of S1 to the Ci-VSP dimer function without breaking the dimer.

Given the variable effects of the S1 mutations on catalysis, we examined the effect of the S1 mutations in the absence of the phosphatase domain. CC1 is a GEVI that has replaced the phosphatase domain with a modified fluorescent protein (FP) to yield a voltage-dependent optical signal (Fig. 7 A) (Piao et al., 2015). This sensor changes fluorescence in a voltage-dependent manner giving a readout of the VSD activation from the cytosolic side. Interestingly, the activation kinetics observed for each mutation in CC1 are significantly different from WT CC1 (Fig. 7 B), unlike the VCF data (Fig. 5 A). While CC1 I131A is slow like CC1 WT, CC1 F127A, I134A, and L137A activate faster than CC1 WT and reach a steady state during the 200-ms pulse (Fig. 7 B). So, while the motions measured from the top of the S4 helix are similar (Fig. 5), the propagated motions through S4 and the linker are different. Similarly, the repolarization kinetics are faster than CC1 WT for CC1 F127A, I134A, and L137A while they are slower than CC1 WT for CC1 I131A. Comparing the steady-state fluorescence versus voltage relationships for these mutations in CC1, we see that both the CC1 L137A and CC1 I131A data appear to be shifted to more negative potentials while the other two mutations, CC1 F127A and CC1 I134A, appear shifted to higher voltages (Fig. 7 C). Since none of the GEVI variants plateaued in response to stronger depolarization steps, the absolute voltage shifts cannot be quantified. However, the relative shifts suggest that both CC1 I131A and CC1 L137A lower the energy needed to move S4, consistent with the VCF data, while CC1 F127A and CC1 I134A require more energy to induce the same change in fluorescence. When we tested CC1 S1-Q, we observed faster kinetics compared with CC1 WT, and the voltage dependence was shifted to lower voltages. These CC1 results indicate that S1 plays a fundamental role in the function of VSDs independent of the phosphatase domain.

## Discussion

We investigated how the S1 helix in VSP impacts Ci-VSP function, from its enzymatic activity to the movements of the VSD, by mutating different amino acids along the S1 helix to alanine. We chose positions that are consistently hydrophobic across several species of VSP (F127, I134, and L137) as well as a position that is not as well conserved (I131). We found that these residues influenced the kinetics and voltage dependence of enzymatic activity even though they do not reside within the phosphatase domain. The same positions also tuned the kinetics and voltage dependence of VSD motions measured in two different ways, VCF and GEVIs. These findings clearly show that the VSP S1 helix is an important consideration for both VSD and enzyme function with multiple possible mechanistic explanations (Fig. 8).

An important point to note is that because VSP is not a channel, we used indirect biosensors to observe the enzymatic activity. Whether the biosensor is a fluorescently dependent sensor as with the PH domains or the biosensor is a PIP-dependent channel, all the measurements are an indirect observation of PIP concentrations. Each type of biosensor has its advantages and disadvantages. Specifically, GFP-PH domains have the disadvantage that they are slow while having the advantage of being able to test all the known reactions catalyzed by VSP. The FRET-based PH domain sensors are faster since the posttranslational modification places the sensors on the membrane. However, as noted by us and others, fPLC does not give a reliable signal for the PI(3,4,5)P$_3$ reaction to PI(4,5)P$_2$ (Rayaprolu et al., 2018; Ratzan et al., 2019; Paixao et al., 2023). Thus, we used the slower gPLC sensor to monitor that reaction. The question arises, why does one biosensor report on the reaction and another does not? We do not expect the absolute binding affinities to be different between the fPLC and gPLC biosensors given that they both use the same PH domain. At the same time, proximity to the membrane will influence the on-and-off kinetics of binding and alter the apparent affinities. Taking this difference

JGP

A

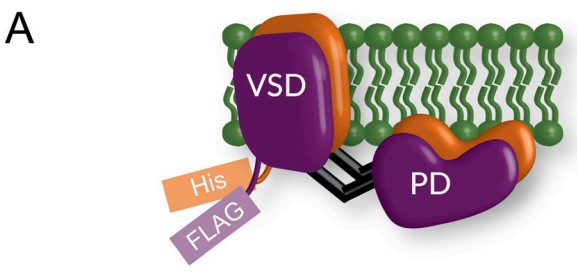

B

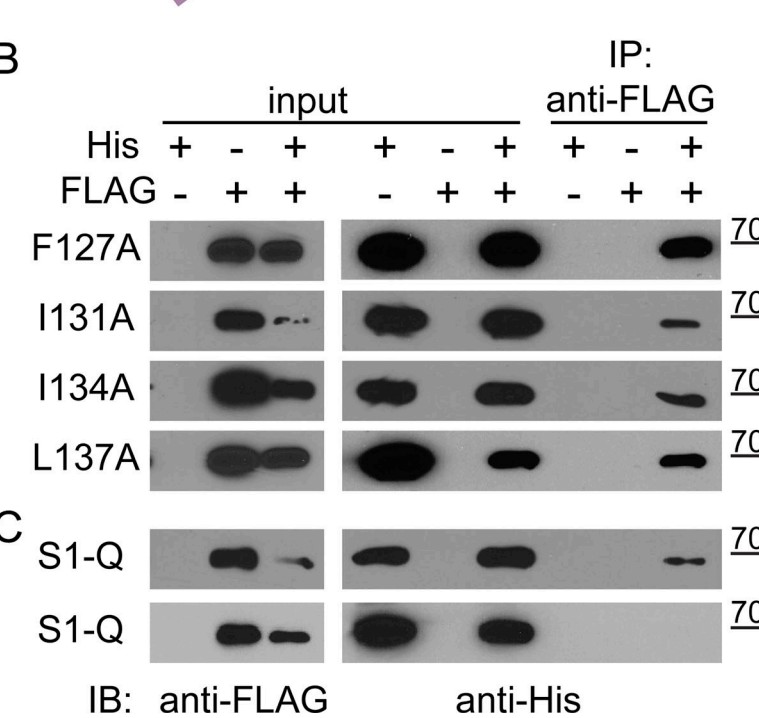

IB:  anti-FLAG        anti-His

**Figure 6.** **Co-IP pull downs show S1 mutations do not disrupt VSP dimers.** **(A)** Cartoon representation of VSP dimers with the His and FLAG tags on the N-terminus. **(B)** Immunoprecipitation assay using individual S1 mutations F127A, I131A, I134A, L137A. Inputs were checked for both FLAG and His to ensure expression. Each blot is representative of at least three pull-down blots and shows the FLAG-tagged subunit is able to pull down the His-tagged subunit. **(C)** Same experiment with all four mutations on a single subunit, called S1-Q for quad mutation on S1. The top blot shows pull down (representative of four separate experiments) while the bottom blot does not (one blot) even though protein levels are similar between the two. Source data are available for this figure: SourceData F6.

into account, we hypothesize that fPLC is at or close to saturated binding to $PI(4,5)P_2$ at typical plasma membrane concentrations. Thus, a cell at rest starts with fPLC already bound to $PI(4,5)P_2$ and lacks the dynamic range to report on the VSP $PI(3,4,5)P_3$ reaction to $PI(4,5)P_2$. Similarly, when we and other groups used channels such as Kir2.1, GIRK, and KCNQ2/3, the rise in $PI(4,5)P_2$ was not observed via a change in the channel current (Villalba-Galea et al., 2009a; Kawanabe et al., 2018; Rjasanow et al., 2015). This lack of a response from the channels could indicate that the channels are already saturated with $PI(4,5)P_2$ and don't respond to the higher concentrations caused by VSP activation. Several other factors could contribute to the discrepancy with channels. Foremost on that list is the starting condition of the cells. It is well known that $PI(3,4,5)P_3$ is at low concentrations in resting cells given its prominent role in regulating cell growth, making any reaction with $PI(3,4,5)P_3$ as a substrate difficult to observe. To make the $PI(3,4,5)P_3$ reactions more observable, we took advantage of the endogenous insulin receptor. We added insulin to our recording solution causing the receptor to activate PI3 kinase, increasing the $PI(3,4,5)P_3$ concentrations. This increases the signal for both the $PI(3,4,5)P_3$ to $PI(4,5)P_2$ reaction and the $PI(3,4,5)P_3$ to $PI(3,4)P_2$ reaction. Other important factors to be considered are the characteristics of the biosensors. PH domains have a one-to-one stoichiometry for binding to PIPs, so cooperativity will not be a factor. For most channels, they bind more

than one PIP and, in several cases, the cooperative nature of those interactions will limit the response of the channel until sufficient $PI(4,5)P_2$ has accumulated. These differences make a cross-comparison between fluorescent biosensors and channels quite difficult. As a result, we focused on comparing the activities of mutations using the same biosensors and comparing them to both positive and negative controls.

With those caveats in mind, we turn to the enzymatic activity where we found that the S1 impacts the enzyme response to voltage. It is important to note that modeling data suggested that the observed voltage dependence of activity using these biosensors is influenced by endogenous non-voltage dependent kinases and phosphatases (Keum et al., 2016; Kruse et al., 2019), which will impact the apparent VSP voltage dependence of activity. However, we were comparing the mutations to our WT protein, so the endogenous proteins are present in all the data and are assumed to contribute the same amount in each. Three of the four individual mutations tested showed conservative changes to enzymatic activity, but all showed some impact depending on the reaction being observed. We separated these differences based on three parameters: the magnitude, the kinetics, and the voltage dependence of the activity. In terms of magnitude, the only major difference was with I134A. More dephosphorylation of $PI(3,4,5)P_3$ was observed and less dephosphorylation of $PI(4,5)P_2$, indicating that the I134A mutation

**Rayaprolu et al.**
Hydrophobic residues in S1 modulate voltage sensing

**Journal of General Physiology**     12 of 17

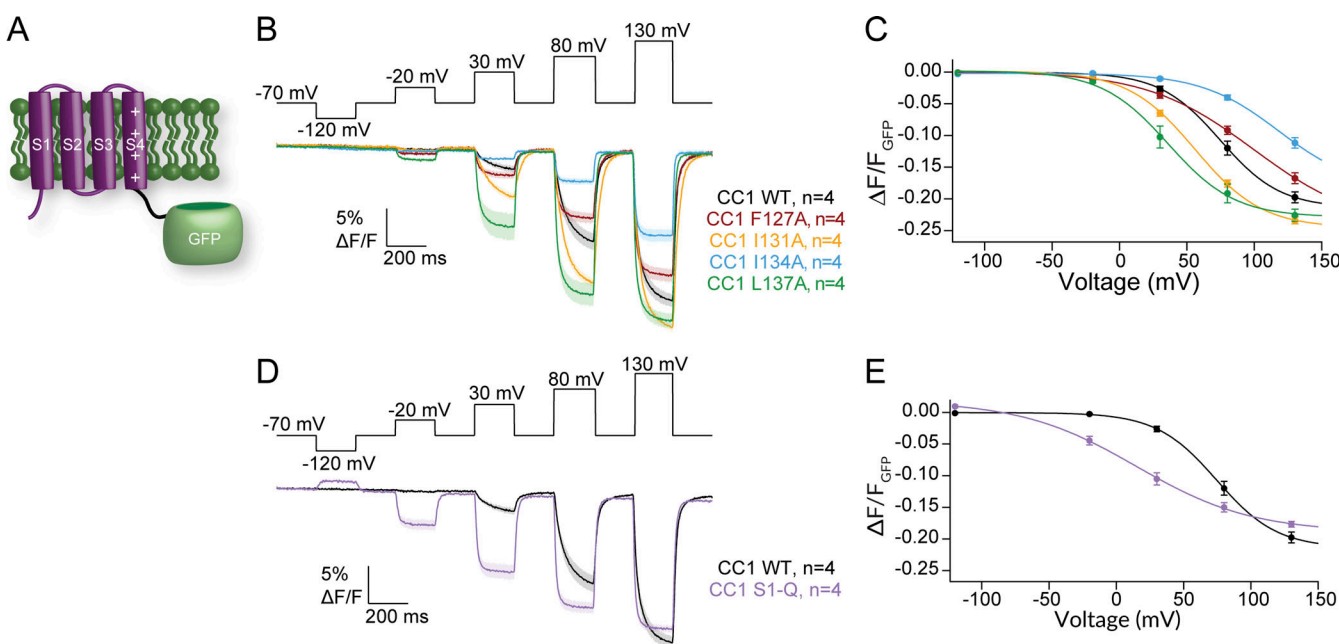

**Figure 7. Impact of S1 mutations on a GEVI. (A)** Cartoon representation of the GEVI CC1 with the VSD from Ci-VSP attached an FP. **(B)** Averaged series of voltage steps showing the kinetics of fluorescence change for CC1 WT and S1 mutants expressed in HEK293 cells. Faster activation and repolarization kinetics observed for F127A, I134A, and L137A. **(C)** Voltage-dependent fluorescence change for WT and S1 mutations. F127A and I134A shift the voltage dependence to higher voltages while I131A and L137A shift to lower voltages. **(D)** Averaged data for a series of voltage steps for WT and S1-Q. S1-Q shows consistently faster kinetics. **(E)** Voltage-dependent fluorescence change for WT and S1-Q. Strong leftward shift consistent with leftward shift from VCF data. All error bars are ± SEM. Data fit with single Boltzmann sigmoid equations.

would leave more PI(4,5)P$_2$ on the membrane compared with WT VSP. Both I131A and I134A showed similar activity magnitudes. For the kinetics, F127A slowed down the PI(3,4,5)P$_3$ to PI(4,5)P$_2$ reaction, and both F127A and I131 slowed the PI(4,5)P$_2$

to PI(4)P$_2$ reactions. The other kinetics were similar to WT. There was more variability in the voltage dependence of these reactions. I131A shifted the PI(3,4,5)P$_3$ to PI(3,4)P$_2$ and PI(3,4)P$_2$ to PI(4)P reactions to higher voltages while the other two

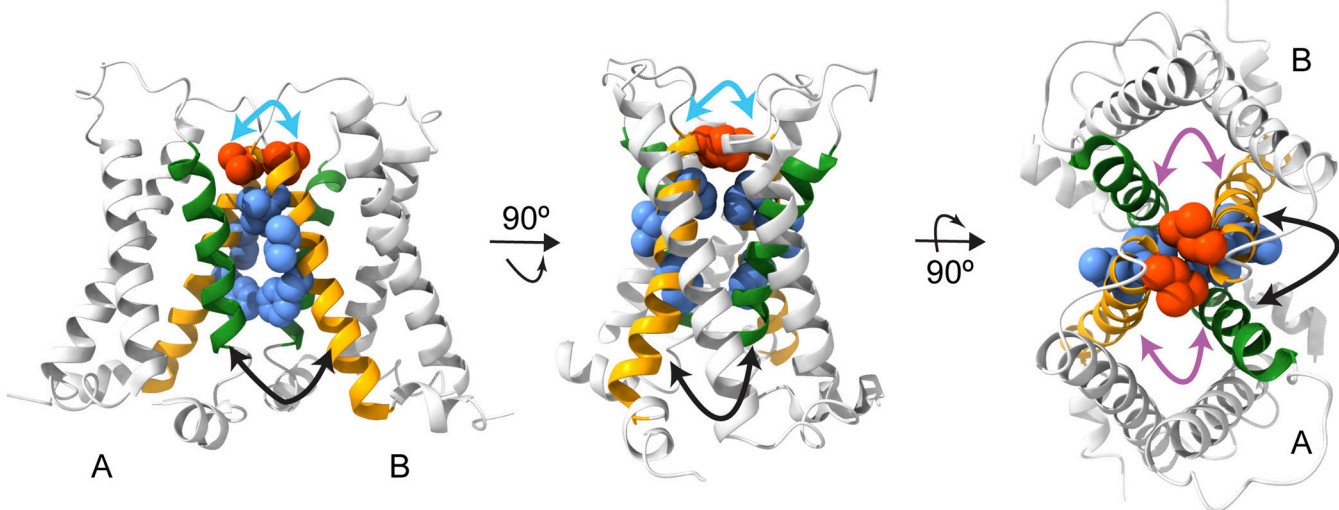

**Figure 8. Model for S1 modulation of S4.** PDB ID 4G80 ribbon rendering of Ci-VSD dimers using ChimeraX (Pettersen et al., 2021). L137 is depicted in red space-filling, F127, I131, and I134 are depicted in blue-space filling, S1 in orange, and S4 in green. Left: View facing into the membrane. S1 from subunit A could be interacting with S4 from subunit B (black arrow) and/or S1 from subunit B (cyan arrow). Middle: Turned-in membrane view. Arrows show the same possible interactions. Right: Top-down view shows how the S1 from one subunit can interact with the S4 from the same subunit (purple arrow) as well as the S4 from the adjacent subunit (black arrow). Cyan arrows indicate possible intermolecular interactions through the S1s at the dimer interface between subunits A and B. Black arrows indicate possible intermolecular interactions between the S1 from subunit B to S4 from subunit A. Purple arrows indicate possible intramolecular interactions between the S1 and S4 of the same subunit.

mutations were similar to WT. For the $PI(3,4,5)P_3$ to $PI(4,5)P_2$ and $PI(4,5)P_2$ to $PI(4)P_2$ reactions, both F127A and I131A were shifted to higher voltages while I134A was similar to WT. While the S4 helix with its charge-carrying arginines is well known to impact VSP enzymatic activity (Murata et al., 2005; Murata and Okamura, 2007; Hossain et al., 2008; Sakata and Okamura, 2014; Mizutani et al., 2022), the role of S1 in VSP is less well understood. These subtle alterations in enzymatic activity from the individual mutations in S1 indicate that hydrophobic residues in S1 transmit information, finely tuning the phosphatase domain activity.

The most dramatic change with the single mutations was with position L137, which is conserved across many different species (Fig. 1 E). By altering the leucine to alanine, the magnitude of activity was consistently reduced, and the voltage dependence was shifted to higher voltages with some appearing more linear (Figs. 3, 4, and 5). The kinetics of activation were also much slower than WT regardless of the biosensor used. These results suggest that the L137 position is a key modulator for VSP and is consistent with its location where L137 is found at the top of each S1 helix and serves as a focal point between the two subunits (Fig. 8). When all four mutations were combined in S1-Q, the kinetics of all the reactions were also reduced in magnitude, slower and shifted to higher energy. While the S1-Q results appear to be dominated by the L137A mutation, there are differences, particularly with the $PI(3,4,5)P_3$ to $PI(4,5)P_2$ reaction. Specifically, the magnitude of the S1-Q activity is more severely reduced and the voltage dependence is also more shifted and more linear compared with the L137A alone (Fig. 4). Thus, the combination of all four mutations in S1 works together to modulate the enzymatic activity of the phosphatase domain. Whether these allosteric impacts are a result of intersubunit interactions or intramolecular interactions within a single subunit remains to be determined.

For the direct measurement of VSD motion via VCF, we found that the activation kinetics of the alanine mutations followed the expected double exponential rates and were mostly similar to WT with minor differences depending on the voltage tested (Table S2; Fig. 5, A and C; and Fig. S3 A) (Kohout et al., 2008; Villalba-Galea et al., 2008). Thus, the alanine mutations did not drastically alter the activation motions of S4. While the activation kinetics are well established to represent the sensing currents ($\tau_{a1}$) and the entry into the VSD relaxed state ($\tau_{a2}$), the repolarization kinetics were always found to go through a single smooth transition from activated to rest ($\tau_{r1}$). F127A, I131A, and I134A showed a single transition with similar rates to WT (Table S2). Interestingly, both the L137A alone and the combination S1-Q introduced a second repolarization transition, $\tau_{r2}$, in addition to the $\tau_{r1}$ transition. Addressing the $\tau_{r1}$ transitions first, the L137A $\tau_{r1}$ transitions were not significantly different from WT across multiple voltages (Fig. S3 B). We did observe a difference between the $\tau_{r1}$ transition between WT and S1-Q at lower voltages but the significance of those differences was not consistent (Fig. S3 B). The $\tau_{r2}$ transitions were similar between L137A alone and S1-Q in that they were not voltage-dependent and were significantly slower than the $\tau_{r1}$ transition. Given these two characteristics, the $\tau_{r2}$ transition could represent a separation

between the sensing charge movement and the transition from relaxed to rest. A voltage-independent return from relaxed to rest would be consistent with previous findings that showed that the transition between the active to the relaxed state was voltage-independent (Villalba-Galea et al., 2008). Another possible interpretation could be that since the $\tau_{r1}$ values are similar between WT and all the single mutations including L137A, the $\tau_{r2}$ transition is due to an additional transition to the sensing charge and relaxed-to-rest transitions, not a separation of the two. Such an additional transition could be due to intersubunit conformational changes between the dimers. Thus, the reason behind the two kinetic repolarization components remains unclear. Further experiments would be needed to explore the additional transition. Next, we plotted the changes in fluorescence against the voltage and found the movement of the mutated positions was separated into two energy transitions, one at low voltage ($V1_{1/2}$) and one at high voltage ($V2_{1/2}$). All the mutations were shifted to lower $V1_{1/2}$ values, indicating less energy was needed for the initial movement of S4 (Fig. 5, B and D; and Table 3). $V2_{1/2}$ was either similar to WT (F127A) or higher than WT (I131A, I134A, L137A, S1-Q), indicating more energy was needed for the second movement of S4. The $V1_{1/2}$ could correspond to one enzymatic reaction and the $V2_{1/2}$ could correspond to another reaction, but the enzymatic activity doesn't support this interpretation. Instead, this separation into low and high energy movements could come from the dimer interactions where each S1 can modulate the S4 from the same subunit and the S4 from the opposing subunit (Fig. 8). If this is the case, the results could indicate that one subunit becomes easier to move and the other subunit becomes harder to move as a result of the S1 mutations. Another possible explanation would be that the alanines impacted the hydrophobic "plug" formed between the S1–S3 helices, separating the energetics of S4 into two motions (Lacroix and Bezanilla, 2012). A third explanation could be the loss of an anchor point. Specifically, alanine mutations could allow S1 to move more freely and no longer serve as a counterpoint to the other helices. Further experiments are needed to separate these possible explanations for the two VSD motions and the lower energy requirements. The movement of S4 was clearly altered based on mutations in S1 indicating the influence S1 has over voltage sensing in VSP.

Interestingly, the GEVI measurement of VSD motion results was quite different than the VCF results. Both CC1 F127A and CC1 I134A required higher voltage and more energy to move the VSD while CC1 I131A, L137A, and S1-Q required less voltage for movement (Fig. 7, C and E). The observed kinetics were also dramatically different. CC1 I131A was similar to CC1 WT while all the other mutations were significantly faster for activation and repolarization (Fig. 7, B and D). This variation from the VCF data could stem from the location of the fluorescence reporter. For VCF, the TMRM is located extracellularly between S3 and S4, reporting the extracellular motions of S4, while for the GEVI, the FP is located intracellularly and so will report on the intracellular motions of S4 through the linker to the FP. The variation could also stem from the fluorophores themselves. TMRM is a small chemical that is very sensitive to its local microenvironment, so as the S4 helix moves, TMRM will move as well, a direct

measure of motion. The FP is an indirect measurement given the voltage change causes a conformational change in S4 which is propagated through the linker to the FP β barrel altering the chromophore and then altering fluorescence. Lastly, the variation could stem from the VSP dimers. While the impact of dimerization is currently unknown for TMRM, previous reports indicate that the dimerization of GEVIs impacts the output fluorescence (Kang and Baker, 2016; Leong et al., 2021). Our results clearly show that the design of GEVIs using VSP VSDs should consider S1 modulation when tuning the GEVI for particular properties.

Previous work analyzed the relaxed state as a mechanism to explain the fluorescence changes in the GEVI VSFP2.3 (Villalba-Galea et al., 2009b). Specifically, VSFP2.3 consists of a tandem pair of FPs fused to the carboxyl end, causing a change in intramolecular FRET when the VSD changes its conformation in response to voltage. The mechanism for CC1 and other ArcLight-derived GEVIs are known to be different. Specifically, there is an intermolecular interaction between the FP domains of neighboring GEVIs. This intermolecular interaction enables the negative charge of the A227D mutation in Super Ecliptic pHluorin to influence the fluorescence of a neighboring chromophore. This interaction of FP domains has been shown by replacing SE with a FRET donor FP and cotransfecting it with a separate GEVI containing a FRET acceptor (Leong et al., 2021). With both domains (the VSD and the FP) experiencing conformational changes, it is difficult to optically resolve the potential relaxation of the VSD from a conformational change relating to the FP domain.

Bringing together the enzymatic and VCF data, we see a complicated picture emerge regarding VSD control of the phosphatase. In particular, we note that the double sigmoid needed to fit the VCF data for all the mutations is not observed in the enzymatic activity data. The lower voltage movements ($V1_{1/2}$) are much lower than the activity voltage dependence, indicating a discrepancy between activation of the VSD and activation of the enzyme. Additionally, the majority of the mutant enzymatic reactions were shifted to higher voltages, indicating more energy was needed to activate the phosphatase with S1 mutated. These opposite impacts could indicate that the ability of the VSD to control the phosphatase was weaker, making it harder for the VSD to conformationally change the phosphatase into its active form, thus requiring higher voltage for catalysis. Decoupling between the VSD and phosphatase has been seen before using mutations in the linker (Kohout et al., 2010; Villalba-Galea et al., 2009a; Hobiger et al., 2012, 2013) and in S4 (Mizutani et al., 2022) but never before from mutations in the S1 helix. The linker decoupling has been long assumed to be an intramolecular process where the VSD no longer communicates with its phosphatase domain. Given the proposed side-by-side organization of the VSP dimer (Fig. 1 C), this interpretation of the data may be oversimplified. In addition, the location of both S1 and S4 at the interface of the VSD dimer (Li et al., 2014) suggests that the observed decoupling with S1 or S4 mutations could be due to either intramolecular interactions as assumed for the linker decoupling or due to intermolecular interactions. Thus, multiple mechanisms are possible for VSP coupling. For example, following a purely intramolecular mechanism, S1 from

subunit A directly interacts with S4 from subunit A with the same happening within subunit B (Fig. 8, purple arrows). In this scenario, the phosphatase domain from subunit A would couple only to the VSD from subunit A. Another possible mechanism could be that the S1 from subunit A could influence the S4 from subunit B (Fig. 8, black arrows). In this case, phosphatase B would receive a signal from VSD B, but the signal would be influenced by VSD A. Similarly, another mechanism could be direct interactions between the S1s mediating the dimer interface (Fig. 8, cyan arrows). This would also lead to a scenario where phosphatase B responds to VSD B but is influenced by VSD A. It is important to note that we are focusing here on the VSD. The phosphatase domains are also known to dimerize (Rayaprolu et al., 2018), so direct interactions between the phosphatases may occur in addition to the VSD interactions. Further experiments will be needed to determine whether the VSD control of the phosphatase is purely intramolecular (intrasubunit), purely intermolecular (intersubunit), or a combination of both.

Overall, our results indicate that the VSP S1 plays an active role in enzymatic function. Specifically, we show that the S1 helix in VSP impacts both the voltage dependence of catalysis as well as the voltage-dependent motions of S4. However, the mechanism behind S1's role is still uncertain. The Hv S1 is believed to mediate both the dimer interface (Hong et al., 2015; Lee et al., 2008; Li et al., 2015) and the cooperativity between the two subunits (Tombola et al., 2010; Gonzalez et al., 2010). The Hv S1 is also involved in channel opening, making it a critical aspect of how Hv functions (Mony et al., 2015). Further experiments are needed to determine whether these aspects of S1 function are true in VSP as well. A better understanding of how S1 helices contribute to VSD function is important for understanding the VSD and VSP as a whole.

### Data availability
The data underlying Fig. 6 are openly available in FigShare at https://doi.org/10.6084/m9.figshare.23960841.

### Acknowledgments
Crina M. Nimigean served as editor.

We thank Y. Okamura for providing the Ci-VSP cDNA and E.Y. Isacoff for the fPLC and fTAPP.

This work was supported by funds from National Institute of General Medical Science of the National Institutes of Health (NIH) grants R01GM111685 (to S.C. Kohout) and P20GM103474 (to G. Mueller, W.D. Baker, M. Fisher, J.T. Kelley, and J.A. Davisson); National Science Foundation grant 2310489 (to S.C. Kohout); Korea Institute of Science and Technology grant 2E32901 (to B.J. Baker). Molecular graphics and analyses performed with the University of California, San Francisco (UCSF) ChimeraX, developed by the Resource for Biocomputing, Visualization, and Informatics at the UCSF, with support from NIH R01GM129325 and the Office of Cyber Infrastructure and Computational Biology, National Institute of Allergy, and Infectious Diseases. Open Access funding provided by Rowan University.

Author contributions: V. Rayaprolu: Data curation, Formal analysis, Investigation, Methodology, H.M. Miettinen: Investigation,

Validation, W.D. Baker: Conceptualization, Formal analysis, Funding acquisition, Investigation, Writing - review & editing, V.C. Young: Formal analysis, Investigation, M. Fisher: Data curation, Formal analysis, Investigation, Visualization, Writing - review & editing, G. Mueller: Investigation, Validation, W.O. Rankin: Formal analysis, Investigation, J.T. Kelley: Formal analysis, Investigation, Software, W.J. Ratzan: Investigation, L.M. Leong: Formal analysis, Investigation, Visualization, J.A. Davisson: Investigation, B.J. Baker: Data curation, Formal analysis, Investigation, Methodology, Writing - original draft, Writing - review & editing, S.C. Kohout: Conceptualization, Data curation, Formal analysis, Funding acquisition, Investigation, Project administration, Supervision, Visualization, Writing - original draft, Writing - review & editing.

Disclosures: The authors declare no competing interests exist.

Submitted: 18 August 2023

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

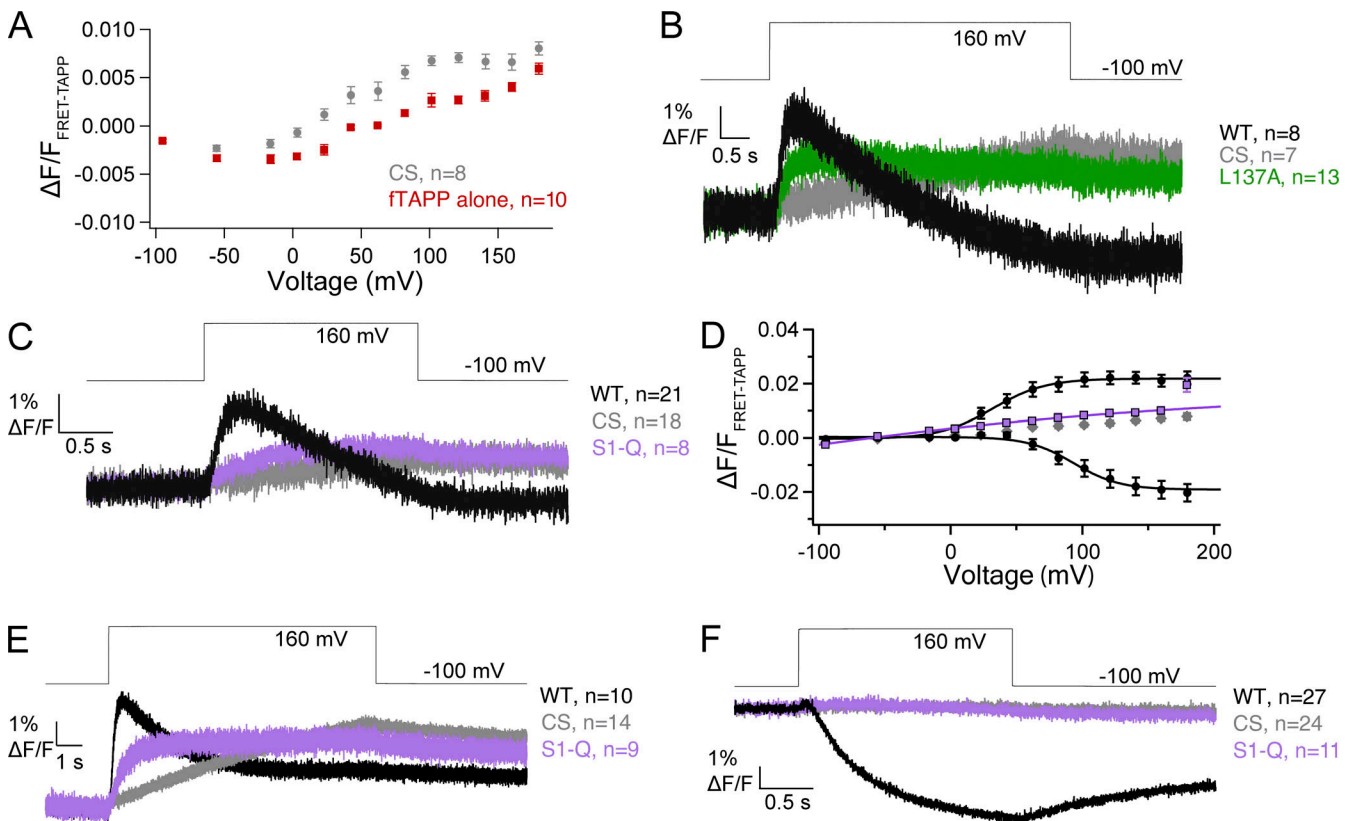

Figure S1. **Impact of oocyte endogenous VSP on heterologous expression of Ci-VSP.** **(A)** Voltage-dependent fTAPP activity in *X. laevis* oocytes. Cells coexpressing catalytically inactive Ci-VSP (C363S, CS) with fTAPP or fTAPP alone. Significantly more background activity from *X. laevis* VSP2 (Xl-VSP2) is observed in cells expressing inactive Ci-VSP. This result suggested that Xl-VSP2 may be more efficiently trafficked to the plasma membrane in the presence of Ci-VSP. **(B)** Averaged data for a long voltage step (5 s) from −100 to 160 mV for L137A, WT, and CS with the fTAPP biosensor. Unsubtracted data from Fig. 3 D. While the CS protein is inactive, the resulting fTAPP increase is significant, indicating a substantial amount of Xl-VSP2 activity for the PI(3,4,5)P$_3$ to PI(3,4)P$_2$ reaction. **(C)** Averaged data for a voltage step from −100 to 160 mV (2 s) for S1-Q. 5-phosphatase activity is dramatically reduced while 3-phosphatase activity appears eliminated. **(D)** Voltage-dependent fTAPP activity for S1-Q, WT, and CS from the 2-s data. While S1-Q is still active, the activity is almost linear and barely above the CS control. **(E)** Averaged data for a long voltage step (10 s) from −100 to 160 mV for S1-Q, WT, and CS with the fTAPP biosensor. The CS protein shows a significant degree of Xl-VSP activity at the longer time scale. Unsubtracted data from Fig. 3 F. **(F)** Averaged data for a 2-s voltage step from −100 to 160 mV for the S1-Q mutation with the fPLC biosensor. S1-Q activity above background (CS) was not observed.

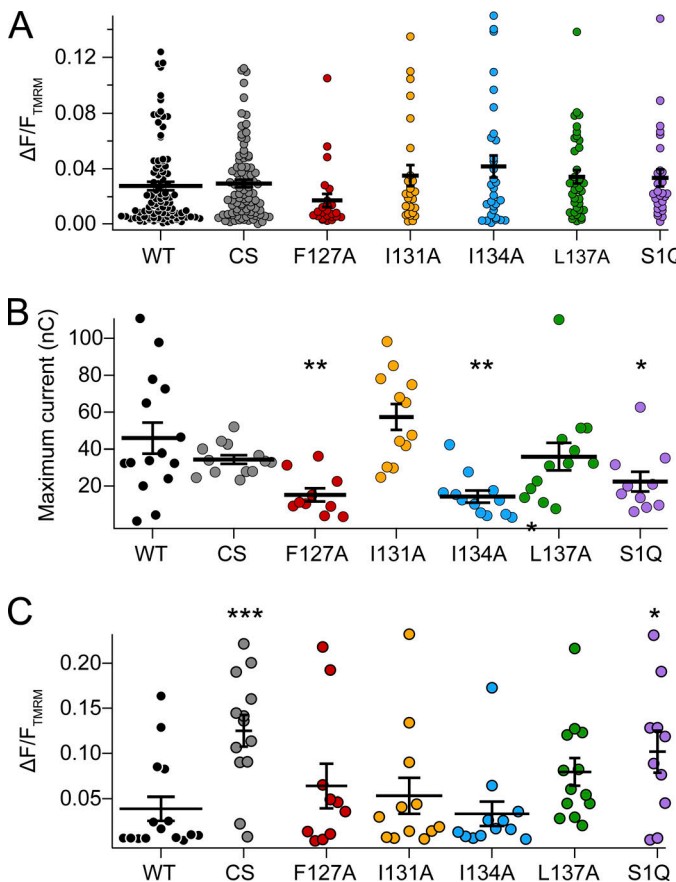

Figure S2. **Quantitation of VSP expression using VCF and sensing currents. (A)** VCF expression data for all the cells used in the activity assays. No statistically significant difference was found between WT and the mutations. WT $n$ = 97, CS $n$ = 95, F127A $n$ = 24, I131A $n$ = 25, I134A $n$ = 31, L137A $n$ = 38, and S1Q $n$ = 27. **(B)** Maximum off-sensing charge from a 150-mV step. Lower current was found for F127A, I134A, and S1-Q. WT $n$ = 15, CS $n$ = 13, F127A $n$ = 10, I131A $n$ = 12, I134A $n$ = 12, L137A $n$ = 13, and S1Q $n$ = 10. **(C)** VCF data for the same cells in B. Larger fluorescence was found for CS and S1-Q. Statistics of mutations versus WT were determined using the Welch's $t$ test with a two-tailed distribution. *** P < 0.001; ** P < 0.01; * P < 0.05.

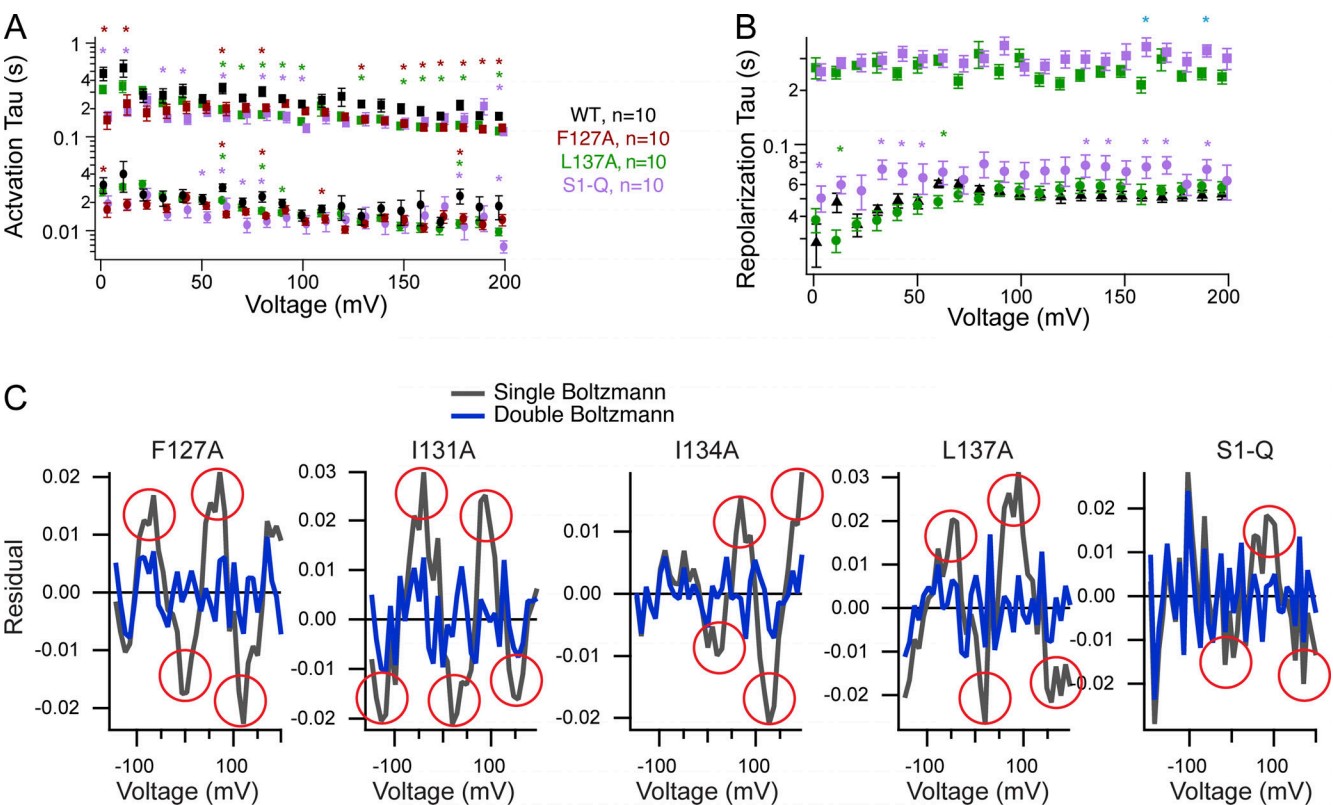

Figure S3.  **Analysis of VCF kinetic and steady-state data. (A)** Activation kinetic analysis for WT, F127A, L137A, and S1-Q VCF. Data from Fig. 6, A and C, were fit with double exponential fits, and the resulting $\tau_{a1}$ and $\tau_{a2}$ values were plotted versus the voltage measured for each trace. Student's *t* tests *P = 0.001–0.049 are shown against WT in the corresponding color for each mutation. **(B)** Repolarization kinetics were fit with a single exponential (WT) or a double exponential (L137A, S1-Q). The resulting $\tau_{r1}$ and $\tau_{r2}$ values were plotted versus the corresponding voltage. Student's *t* tests *P = 0.0025–0.034 are shown against WT in the corresponding color for each mutation, except for $\tau_{r2}$ values which are shown in cyan and are L137A versus S1-Q. **(C)** Residual for each of the individual mutations and S1-Q, comparing the single and double fits. Significant discrepancies in the single Boltzmann sigmoid are circled in red. The double Boltzmann sigmoid consistently fits all the mutant VCF data better (fits from Fig. 5, B and D).

**Provided online are two tables. Table S1 lists the primers used in this study. Table S2 lists VCF kinetics of S4 motions from the 200-mV step.**

