## [Peer Review File · The Journal of General Physiology]

Hydrophobic residues in S1 modulate enzymatic function and voltage sensing in voltage sensing phosphatase

Vamseedhar Rayaprolu, Heini Miettinen, William Baker, Victoria Young, Matthew Fisher, Gwendolyn Mueller, William Rankin, John Kelley, William Ratzan, Lee Leong, Joshua Davisson, Bradley Baker, and Susy Kohout

Corresponding Author(s): Susy Kohout, Cooper Medical School of Rowan University

Review Timeline:

Submission Date:	August 18, 2023
Editorial Decision:	September 21, 2023
Revision Received:	December 13, 2023
Editorial Decision:	January 11, 2024
Revision Received:	April 8, 2024
Editorial Decision:	April 30, 2024
Revision Received:	May 3, 2024

Editor: Crina Nimigean

Transaction Report:

DOI: <https://doi.org/10.1085/jgp.202313467>

September 20, 2023

Dr. Susy C. Kohout
Cooper Medical School of Rowan University
Biomedical Science
401 South Broadway
Camden, New Jersey 08103

Re: 202313467

Dear Susy,

Your manuscript entitled "Impact of S1 hydrophobic residues on voltage sensing" has now been seen by 3 reviewers, whose comments are appended below. You will see that although both the editors and the reviewers appreciated the significance of addressing this previously unexamined aspect of the voltage sensor, serious concerns were raised about the manuscript. Importantly, while the experiments indeed revealed that the S1 segment contributes in some fashion to the voltage sensor and the coupled phosphatase function, the mechanism behind it was not clear, the experiments appeared rather unfocused and did not appear to test a specific mechanistic hypothesis.

For example, it was not clear whether the role of the dimerization interface was meant to be probed with the chosen mutations. If yes, the rationale for the mutations is unclear, and if not, it is unclear what was the objective. Moreover, the colP experiments only reveal the existence of some dimers at the surface containing both tags and do not rule out partially impaired dimerization that could still lead to the observed effects of the S1 mutants on the VSD and phosphatase. More experiments and controls are needed to rule out dimerization as a culprit for the observed effects. In addition, the observed effects of the mutations on the phosphatase function and voltage sensor activity are complex and, even if the dimerization is not impaired at all, there are, perhaps understandably, limited insights on the role of S1 on either of those domains and on the coupling between the VSD and the enzyme. Thus, the conclusions from this study as presented remain limited and the manuscript appears disjointed despite the large amount of data provided.

As a result of these concerns, I am afraid that we cannot offer to publish your manuscript. However, if you believe that further experiments would allow you to address these issues to the satisfaction of the editors and reviewers, we would be willing to consider a substantially revised manuscript if resubmitted within one year. If you choose to do so, we recommend that you outline clearly the hypotheses to be tested and focus considerably the manuscript for the revision.

In the case of eventual publication, the article would include a 'revised date' alongside its submitted and accepted dates. In the event that the revision process takes longer, any resubmitted manuscript would be treated as a new submission and would be given a new manuscript number. Either way, any revised manuscript would be sent out for review to the original reviewers, subject to their availability, and we must emphasize that we cannot predict the final outcome.

Should you decide to submit a revised version, please submit your revised manuscript via the link below and include a letter that details your responses to the editors' and reviewers' concerns. Also, to facilitate evaluation of revised manuscript by Reviewers and Editors, please provide a copy of the text with alterations highlighted (e.g., boldfaced, underlined, or in color). If you feel changes are too extensive to provide a version of your paper with changes highlighted, please state so in your response.

Please pay particular attention to recent changes to our instructions to authors in the following sections: Data presentation, Blinding and randomization and Statistical analysis, under Materials and Methods, as shown here: <https://rupress.org/jgp/pages/submission-guidelines#prepare>. Re-review will be contingent on inclusion of the required information (including for data added during revision) and demonstration of the experimental reproducibility of the results. Also, to improve the reproducibility of published content, we have partnered with SciScore. Authors are prompted in eJP to copy and paste the Materials and Methods section of their manuscript for a SciScore assessment when submitting their revised manuscript. Authors are encouraged (not required) to further revise their Materials and Methods if the SciScore is below 4. More information can be found here: <https://rupress.org/jgp/pages/submission-guidelines#sciscore>

Please do not hesitate to contact me (via the editorial office) if you feel that a discussion of the reviewers' and editors' comments would be helpful.

Thank you for the opportunity to consider your manuscript, a copy of which we will retain for our files.

Sincerely,

Journal of General Physiology's mission is to publish mechanistic and quantitative molecular and cellular physiology of the highest quality; to provide a best-in-class author experience; and to nurture future generations of independent researchers.

Please submit your revised manuscript via this link within one year:
Link Not Available

Reviewer #1 (Comments to the Authors):

The manuscript titled "Impact of S1 hydrophobic residues on voltage sensing" by Rayaprolu and colleagues describe the findings from two teams of investigators, the Bradley lab and the Kohout lab, investigating the roles of a group of hydrophobic residues located S1 segment of the *Ciona intestinalis* Voltage Sensitive Phosphatase (Ci-VSP). To do so, the researchers mutated these residues into alanine, and assessed the effect of each mutation on enzymatic activity and voltage-dependent movement of the voltage-sensing domain (VSD) of this protein. The work described in the manuscript provides new information on the role of the S1 segment in voltage sensing, which has been understudied for this protein. Thus, this work constitutes an important contribution to the field. However, there are some major concerns that need to be addressed before the manuscript can be published.

Major concerns:

1. Page 8, second paragraph: The authors normalized the curves for activity as a function of the potential, using the activity of the endogenous VSP as a reference. It is not clear what is the rationale behind normalizing this signal. The effect of the endogenous VSP - which seems negligibly for potentials below +100 mV - is an additive component to the overall signal. Thus, subtraction rather than normalization would be a more proper approach to account for this background signal.
2. Page 9, second paragraph: The data in Fig 5 data suggest that, upon activation of the VSP, there is an increase in the plasma membrane PI(4,5)P2 from PI(3,4,5)P3, if this is the case, co-expression of Ci-VSP with either a Kir channel or KV7.2/KV7.3 channel will result in an increase or not change in the activity of these channels. However, many publications show the opposite. Please explain your results considering these previous observations.
3. Page 9, third paragraph, top: It has been shown that the fluorescent signal from TMRM attached to the residue G214C reports two distinct processes in the dynamics of the VSD (PMID: 18818307). The first phase of the signal is associated with the movements of sensing charges in the VSD, as it is directly correlated to sensing currents. The second (late) phase of the fluorescent signal is correlated to VSD relaxation. During this process the voltage dependence of charge movement is shifted to more negative potentials - this also has been shown for GEVI (PMID: 19167283). Also, this indicates that the deactivation of the VSD becomes slower as the displacement of sensing charges occurs at more negative potentials. In this study, the enzymatic activity of the mutant L137A was shown to be shifted to more positive potentials respect to that of the WT. In Figure 6, the effect has the opposite trend. Could it be possible that the VSD relaxation observed with the mutant L137A is "deeper"/"stronger" than the relaxation observed with the WT? So, what is being observed is the voltage dependence of the relaxed state?
4. Page 9, third paragraph, bottom: The fluorescent emission traces shown in Figure 6A strongly suggest that the deactivation becomes slower for the mutant L137A. This suggests that the mutant VSP is more resilient to deactivate. So, a more negative potential must be needed to deactivate the VSD at a similar rate than that of the WT. This is consistent with a shift in voltage dependence towards more negative potentials, like what happens following VSD relaxation. As the voltage dependence and kinetics are tightly related, it would be very informative to see how the time constants of activation and deactivation behave as a function of the activating potential.
5. Page 10, first paragraph: Like the previous case with TMRM (Fig. 6), the emission from the fluorescent protein attached to the C-terminal of Ci-VSP can be modulated by conformational changes in the VSD. However, there are at least two voltage-driven changes, one directly associated with sensing charge movements (current) and those subsequent ones associated with VSD relaxation (PMID: 19167283). In this case, it is not clear which of the two changes are reported by the GFP, if not both. The authors should consider this alternative interpretation of their observation.
6. Page 11, third paragraph: Same concern as on #3 and #5. Particularly, the two components in the movement of the VSD as reported by TMRM fluorescent emission have been correlated to sensing charge movement (fast) and to VSD relaxation (slow). This is also the case for GEVI.
7. Page 12, second paragraph: The authors stated "Both CC1 F127A and CC1 I134A required higher voltage, more energy, to move the VSD while CC1 I131A, L137A and S1-Q all required less voltage for movement (Fig 7C,E)." This conclusion appears to disregard that the kinetics of the signals for F127A and I134A seem to plateau, while the other mutants are not. This means that further changes in $\Delta F/F$ could be expected, so underestimating the maximum $\Delta F/F$ in the emission. This will likely cause a shift to more negative potential in the $\Delta F/F$ vs V curves.

Minor concerns:

1. Page 8, top: The authors should consider acknowledging that the residue D136 has been shown to produce a strong shift toward more positive potentials in the activity of Ci-VSP (PMID: 22896666).

Reviewer #2 (Comments to the Authors):

This study investigates the role of the S1 transmembrane helix in the function of the voltage sensitive phosphatase CiVSP. Alanine mutations were introduced at selected positions within S1, individually or in combination, and the function of the resulting mutant proteins were examined by two fluorescence-based approaches: one reporting on enzymatic activity, the other reporting on conformational changes in the voltage sensing domain. Furthermore, corresponding S1 mutations were introduced in CC1, a genetically encoded voltage indicator based on the VSD of CiVSP linked to a fluorescent protein, to test their effects on size, kinetics, and voltage dependence of the fluorescence signal.

The approaches are rigorous and the main conclusion, that S1 is important for influencing the function of the VSD and for tuning the activity of the coupled enzyme is well supported. Yet, the manuscript would benefit from a clearer introduction of the mechanistic hypothesis that the study was meant to test. An initial goal might have been the separation of the two subunits in the dimer, but that did not work. In that case, why not introducing more disruptive mutations like bulky tryptophanes, as previously done by the Miller group for the S1-S3 helices of the Shaker VSD (Monks et al. JGP 1999, Hong & Miller JGP 2000) or hydrophilic/charged residues? If the goal was to explore the role of the dimer interface without breaking the dimer, why limiting the analysis only to positions predicted to be at the interface without controls outside the interface?

The conclusion section also needs clarification. It is mostly focused on the general importance of S1 on function, but for that, one could have limited the analysis to the enzymatic activity. The point of investigating the effects of the mutations on the movement of the voltage sensor, and extending the analysis to CC1, is kind of lost in the conclusion. The authors present numerous possible interpretations of the data but do not offer much mechanistic insight on the coupling between sensor and phosphatase domain. Granted that the observed effects of the mutations are complex and will need further studies to be completely elucidated. Still, the author can focus on discussing the basic role of S1 on modulating the voltage sensor and possibly its allosteric coupling to the enzyme. The idea that the mutations decoupled the VSD from the enzyme is interesting and should be discussed further in terms of how it could happen.

Minor Points

Page 10: (F127, I134 and L137A) should be L137

Page 12: Incomplete/unclear sentence "This suggests that the S1 in VSP also plays an active role in function as the S1."

Reviewer #3 (Comments to the Authors):

This study aims to address the previously unappreciated impact of S1 in the voltage sensing domain from VSP. While the topic itself is of general significance, the work is fragmented and many gaps are filled with speculation. Their conclusion that the S1 helix of VSP plays an important role, while probably true, does not seem to provide more knowledge. The mechanistic aspects of how S1 plays an important role are not sufficiently resolved.

Focusing on the potential dimer interface at S1 found in the crystal structure (4G80), authors designed five different VSP mutants and characterized in terms of dimerization (Fig. 2), phosphatase activities (Fig. 3-5), VSD motion (Fig. 6) and application to GEVI (Fig. 7). Given that effect on dimerization is unclear (Fig. 2), it is difficult to interpret the altered voltage dependency and phosphatase activities through the breakdown of dimerization. Also, the application to GEVI does not seem to contribute much to the present GEVI field.

1. To evaluate surface expression level, Q/V curves of sensing currents should be at least shown for the mutant analyzed. This is necessary to rule out the possibility that the right-shifted phosphatase activity shown later is not due to reduced surface expression.
2. The Co-IP may support existence of dimer but does not exclude VSP monomer at the membrane. Then how is it possible to discriminate the contribution from these two in the later characterizations? Also, details of the constructs including tag locations should be presented.
3. Analysis of the F-TAPP and F-PLC signals is only superficially done by plotting peak values against voltage (Fig 3, 4). Is it not possible, for example, to analyze activation kinetics with numerical modeling?
4. Ref #11 (Lacroix et al., 2012) have previously demonstrated importance of S1 through the effect of side chain surface at I126. Isn't there a similar effect in the present mutants?
5. The optical signal of the S1-Q VCF (Fig 6C) seems to be noisier than the WT. Does this reflect a decrease in response amplitude? If this difference is significant, what does it mean?
6. Authors should explain rationale for the idea that XI-VSP2 protein contributes to the endogenous voltage dependent activity of increase of PI(3,4)P2 in *Xenopus* oocyte. It is not clear.
7. There seems to be a typo "H" in Fig. 3F.

Rayaprolu et al- Response to Reviewer Comments

We thank all the reviewers for their constructive comments to help us improve this manuscript. Below are our responses (black) to specific comments (blue). We did not include quotations of the revised text in our response here because the revisions were extensive. However, we highlighted them in green in the manuscript as we made them to make it easier to focus on the parts that changed.

Reviewer #1 (Comments to the Authors):

The manuscript titled "Impact of S1 hydrophobic residues on voltage sensing" by Rayaprolu and colleagues describe the findings from two teams of investigators, the Bradley lab and the Kohout lab, investigating the roles of a group of hydrophobic residues located S1 segment of the *Ciona intestinalis* Voltage Sensitive Phosphatase (Ci-VSP). To do so, the researchers mutated these residues into alanine, and assessed the effect of each mutation on enzymatic activity and voltage-dependent movement of the voltage-sensing domain (VSD) of this protein. The work described in the manuscript provides new information on the role of the S1 segment in voltage sensing, which has been understudied for this protein. Thus, this work constitutes an important contribution to the field. However, there are some major concerns that need to be addressed before the manuscript can be published.

Major concerns:

1. Page 8, second paragraph: The authors normalized the curves for activity as a function of the potential, using the activity of the endogenous VSP as a reference. It is not clear what is the rationale behind normalizing this signal. The effect of the endogenous VSP - which seems negligibly for potentials below +100 mV - is an additive component to the overall signal. Thus, subtraction rather than normalization would be a more proper approach to account for this background signal.

We agree with the reviewer that a subtraction is more appropriate than a normalization. We apologize for using the incorrect word. In the Methods section, under Data analysis, we describe our calculations and state that we subtracted the CS from WT and from the mutations. The word "normalization" in the results section has been removed and "subtracted" is used instead.

2. Page 9, second paragraph: The data in Fig 5 data suggest that, upon activation of the VSP, there is an increase in the plasma membrane PI(4,5)P₂ from PI(3,4,5)P₃, if this is the case, co-expression of Ci-VSP with either a Kir channel or KV7.2/KV7.3 channel will result in an increase or not change in the activity of these channels. However, many publications show the opposite. Please explain your results considering these previous observations.

The reviewer correctly points out the limitations of different biosensors when studying PIP concentrations. The characteristics of the biosensor, whether a PH domain or a channel, will dictate how the signal appears. Channels typically require more than one PI(4,5)P₂ for activation while the PH domains are a one to one interaction. This stoichiometry difference will influence how the channels respond to the changing PIP concentrations relative to the PH domains. In addition, the low resting concentration PI(3,4,5)P₃ will impact the ability to observe dephosphorylation events from PI(3,4,5)P₃ as a substrate. A discussion of this has been added to manuscript.

3. Page 9, third paragraph, top: It has been shown that the fluorescent signal from TMRM attached to the residue G214C reports two distinct processes in the dynamics of the VSD (PMID: 18818307). The first phase of the signal is associated with the movements of sensing charges in the VSD, as it is directly correlated to sensing currents. The second (late) phase of the fluorescent signal is correlated to VSD relaxation. During this process the voltage

dependence of charge movement is shifted to more negative potentials - this also has been shown for GEVI (PMID: 19167283). Also, this indicates that the deactivation of the VSD becomes slower as the displacement of sensing charges occurs at more negative potentials. In this study, the enzymatic activity of the mutant L137A was shown to be shifted to more positive potentials respect to that of the WT. In Figure 6, the effect has the opposite trend. Could it be possible that the VSD relaxation observed with the mutant L137A is "deeper"/"stronger" than the relaxation observed with the WT? So, what is being observed is the voltage dependence of the relaxed state?

The reviewer brings up an interesting point regarding the relaxed state with these mutations. As pointed out by the reviewer, the first phase is correlated with the sensing charge movement and we did not see significant changes between WT and the single mutations with this first rate (Supp Table 1). However, we did observe some differences in the second rate that correlates with the relaxed state so we conducted a more extensive analysis across multiple voltages. We found that both F127A and L137A entered the relaxed state slightly faster than WT at some voltages though not all the differences were significant (see new supplemental figure 2A). The voltage dependence of either rate did not differ between WT and L137A or WT and F127A suggesting that neither F127A nor L137A was entering a deeper relaxed state. We also analyzed the repolarization kinetics for WT, L137A and S1Q. The first rate (only rate for WT, τ_{r1}) were similar for both WT and L137A. The second rate, τ_{r2} , was found to be voltage independent which suggests an energy independent extra transition could be needed for L137A to leave the relaxed state or could represent a separation between the sensing charge movement and the transition from relaxed to rest. Interestingly, the same analysis for S1Q found that τ_{r1} differed at lower voltages making the overall voltage dependence of the first rates less dependent on voltage however the differences were not consistent (new supplemental Figure 2B). The second rate, τ_{r2} , was as voltage independent as L137A. This extra analysis does not clearly favor an extra transition leaving the relaxed state versus a separation of the sensing charge movement from the transition from relaxed to rest. These points are important as brought up by the reviewer so we added this data to the results as well as into the discussion.

4. Page 9, third paragraph, bottom: The fluorescent emission traces shown in Figure 6A strongly suggest that the deactivation becomes slower for the mutant L137A. This suggests that the mutant VSP is more resilient to deactivate. So, a more negative potential must be needed to deactivate the VSD at a similar rate than that of the WT. This is consistent with a shift in voltage dependence towards more negative potentials, like what happens following VSD relaxation. As the voltage dependence and kinetics are tightly related, it would be very informative to see how the time constants of activation and deactivation behave as a function of the activating potential.

We agree and have added that analysis as per point 3.

5. Page 10, first paragraph: Like the previous case with TMRM (Fig. 6), the emission from the fluorescent protein attached to the C-terminal of Ci-VSP can be modulated by conformational changes in the VSD. However, there are at least two voltage-driven changes, one directly associated with sensing charge movements (current) and those subsequent ones associated with VSD relaxation (PMID: 19167283). In this case, it is not clear which of the two changes are reported by the GFP, if not both. The authors should consider this alternative interpretation of their observation.

The reviewer points out an important point. The mechanism(s) mediating fluorescence change for the GEVI VSFP2.3 analyzed nicely in Villalba-Galea et al., 2009 is distinct from the GEVIs described here. VSFP2.3 consists of a tandem pair of FPs fused to the carboxyl end causing a change in intramolecular FRET when the VSD changes its conformation in response to voltage. For CC1 and other ArcLight-derived GEVIs, there is an intermolecular interaction between the FP domains of neighboring GEVIs. This intermolecular interaction enables the negative charge of the A227D mutation in Super Ecliptic

pHluorin to influence the fluorescence of a neighboring chromophore. This interaction of FP domains has been shown by replacing SE with a FRET donor FP and cotransfecting it with a separate GEVI containing a FRET acceptor (Leong et al., 2021). Please see Figure 1 in Leong et al, 2021.

The location of the external charge in the FP domain has a dynamic effect of the voltage-dependent optical signal (Kang et al., 2021). Moving the negative charge to different external positions can yield multicomponent or inverted optical signals. Please see Figures 2 and 4 in Kang et al, 2021.

A Crystal Structure of GFP

B TM Aspartic Acid Scan

In addition, we have evidence that a mutation to an internal side chain in the FP domain near the chromophore alters the kinetics of the voltage-dependent optical signal indicating that the FP domain also experiences a conformational change when the VSD moves. With both domains (the VSD and the FP) experiencing conformational changes, it is difficult to optically resolve the potential relaxation of the

VSD from a conformational change relating to the FP domain. We added text in the discussion to address this point.

6. Page 11, third paragraph: Same concern as on #3 and #5. Particularly, the two components in the movement of the VSD as reported by TMRM fluorescent emission have been correlated to sensing charge movement (fast) and to VSD relaxation (slow). This is also the case for GEVI.

We agree and we addressed it in point 3, in the results and in the discussion.

7. Page 12, second paragraph: The authors stated "Both CC1 F127A and CC1 I134A required higher voltage, more energy, to move the VSD while CC1 I131A, L137A and S1-Q all required less voltage for movement (Fig 7C,E)." This conclusion appears to disregard that the kinetics of the signals for F127A and I134A seem to plateau, while the other mutants are not. This means that further changes in $\Delta F/F$ could be expected, so underestimating the maximum $\Delta F/F$ in the emission. This will likely cause a shift to more negative potential in the $\Delta F/F$ vs V curves.

The reviewer is correct in noting that the F127A, I134A, and CC1 do not plateau causing an underestimation of the optical signal. But the error of this underestimation likely shifts the voltage response to more negative potentials. For instance, the slope of the change for I131A is larger for the 200 mV depolarization than the 150 or 100 mV steps. In addition, S1-Q GEVI responds to hyperpolarization steps of the plasma membrane while CC1 clearly does not. Another issue is that our setup is limited to 5000 frames each recording. Extending the pulse length is possible but then we cannot see all pulses in the same trial.

Minor concerns:

1. Page 8, top: The authors should consider acknowledging that the residue D136 has been shown to produce a strong shift toward more positive potentials in the activity of Ci-VSP (PMID: 22896666).

We thank the reviewer for pointing out this mutation which we missed in our literature review for this manuscript! A discussion is now included.

Reviewer #2 (Comments to the Authors):Reviewer #2 (Comments to the Authors):
This study investigates the role of the S1 transmembrane helix in the function of the voltage sensitive phosphatase CiVSP. Alanine mutations were introduced at selected positions within S1, individually or in combination, and the function of the resulting mutant proteins were examined by two fluorescence-based approaches: one reporting on enzymatic activity, the other reporting on conformational changes in the voltage sensing domain. Furthermore, corresponding S1 mutations were introduced in CC1, a genetically encoded voltage indicator based on the VSD of CiVSP linked to a fluorescent protein, to test their effects on size, kinetics, and voltage dependence of the fluorescence signal.

The approaches are rigorous and the main conclusion, that S1 is important for influencing the function of the VSD and for tuning the activity of the coupled enzyme is well supported. Yet, the manuscript would benefit from a clearer introduction of the mechanistic hypothesis that the study was meant to test. An initial goal might have been the separation of the two subunits in the dimer, but that did not work. In that case, why not introducing more disruptive mutations like bulky tryptophanes, as previously done by the Miller group for the S1-S3 helices of the Shaker VSD (Monks et al. JGP 1999, Hong & Miller JGP 2000) or hydrophilic/charged residues? If the goal was to explore the role of the dimer interface without breaking the dimer, why limiting the analysis only to positions predicted to be at the interface without controls outside the interface?

We agree with the reviewer that a bulky tryptophan could break the dimer interface. The purpose of this manuscript was to explore how positions in S1 that could mediate the interface will influence the function of the protein, both in VSD motions and enzymatic activity. A manuscript depicting the monomerization of VSP is separate from what our goal is here. We tightened up the introduction and several other areas where our goals may not have been clear.

The conclusion section also needs clarification. It is mostly focused on the general importance of S1 on function, but for that, one could have limited the analysis to the enzymatic activity. The point of investigating the effects of the mutations on the movement of the voltage sensor, and extending the analysis to CC1, is kind of lost in the conclusion. The authors present numerous possible interpretations of the data but do not offer much mechanistic insight on the coupling between sensor and phosphatase domain. Granted that the observed effects of the mutations are complex and will need further studies to be completely elucidated. Still, the author can focus on discussing the basic role of S1 on modulating the voltage sensor and possibly its allosteric coupling to the enzyme. The idea that the mutations decoupled the VSD from the enzyme is interesting and should be discussed further in terms of how it could happen.

The actual mechanism of coupling between the VSD and the phosphatase domain has been the subject of many different articles. We include more discussion on the insights provided by these data though as the reviewer notes, further studies will be needed to truly elucidate the mechanism.

Minor Points

Page 10: (F127, I134 and L137A) should be L137

We corrected the typo. Thank you for pointing it out!

Page 12: Incomplete/unclear sentence "This suggests that the S1 in VSP also plays an active role in function as the S1."

We corrected the incomplete sentence. Thank you for pointing it out!

Reviewer #3 (Comments to the Authors):

This study aims to address the previously unappreciated impact of S1 in the voltage sensing domain from VSP. While the topic itself is of general significance, the work is fragmented and many gaps are filled with speculation. Their conclusion that the S1 helix of VSP plays an important role, while probably true, does not seem to provide more knowledge. The mechanistic aspects of how S1 plays an important role are not sufficiently resolved.

Focusing on the potential dimer interface at S1 found in the crystal structure (4G80), authors designed five different VSP mutants and characterized in terms of dimerization (Fig. 2), phosphatase activities (Fig. 3-5), VSD motion (Fig. 6) and application to GEVI (Fig. 7). Given that effect on dimerization is unclear (Fig. 2), it is difficult to interpret the altered voltage dependency and phosphatase activities through the breakdown of dimerization. Also, the application to GEVI does not seem to contribute much to the present GEVI field.

1. To evaluate surface expression level, Q/V curves of sensing currents should be at least shown for the mutant analyzed. This is necessary to rule out the possibility that the right-shifted phosphatase activity shown later is not due to reduced surface expression.

We agree that expression will directly impact the outcome of the enzymatic activity assays. As such, for each cell used for an activity assay, we made sure the cells were expressing VSP by testing their VSD

motions through VCF. In this context, the VCF serves the same purpose as the sensing current measurement. We clarified in the methods section that this control was conducted.

2. The Co-IP may support existence of dimer but does not exclude VSP monomer at the membrane. Then how is it possible to discriminate the contribution from these two in the later characterizations? Also, details of the constructs including tag locations should be presented.

We agree with the review that both monomers and dimers are likely present at the concentrations we are using. This assumption is consistent with previous publications. In terms of discriminating between the two, without a clear way of monomerizing the dimers, we do not believe there's a clear way of separating the two. Monomerizing the dimers is beyond the scope of this publication. We added a cartoon depicting the tag locations.

3. Analysis of the F-TAPP and F-PLC signals is only superficially done by plotting peak values against voltage (Fig 3, 4). Is it not possible, for example, to analyze activation kinetics with numerical modeling?

We agree with the reviewer that modeling of the kinetics would be an interesting exercise. The modeling would need to include several parameters that we are unable to determine empirically including the binding affinities of the fTAPP and fPLC biosensors. Their proximity to the membrane means they will have different kinetics of binding relative to previous studies that modeled the diffusion based biosensors. A full model these FRET based biosensors would be needed first. Thus, we believe the suggested modeling experiment are beyond the scope of this manuscript.

4. Ref #11 (Lacroix et al., 2012) have previously demonstrated importance of S1 through the effect of side chain surface at I126. Isn't there a similar effect in the present mutants?

The reviewer poses an interesting question. The previous study of Lacroix et al certainly shed light on the importance of amino acid size in the S1 of the Ci-VSD by changing the same position to all the amino acids. While a survey of all the amino acids in positions 127, 131, 134 and 137 is beyond the scope of this manuscript, the impacts we observe are likely due to the smaller size of the Ala. Our intention was to probe how these residues impact not only VSD function but also enzymatic function. Lacroix et al do not investigate enzymatic function so whether the effect is similar is not clear. The equivalent position was mutated to a Phe in Dr-VSP and the activity appeared shifted in the same, lower voltage direction (Sakata and Okamura, 2014). The mutations we tested generally shift in opposite directions, with the VSD easier to move and the phosphatase harder to activate. This would suggest the effects are not similar. However many more experiments would need to be done to address whether the effects were similar between the data in Lacroix et al 2012 and the data we present here.

5. The optical signal of the S1-Q VCF (Fig 6C) seems to be noisier than the WT. Does this reflect a decrease in response amplitude? If this difference is significant, what does it mean?

The reviewer is correct that the larger noise in the representative traces are likely correlated with a lower overall amplitude change. The amplitude and the resulting noise are dependent on several factors including protein expression levels and the PMT voltage. The PMT voltages are not correlated with protein expression levels since they are not linear in their response to photons. While it is tempting to interpret the relative amplitude in VCF as meaningful, the mechanism behind the VCF signal is unknown. For example, position G214C gives a fluorescence decrease while Q208C in the same protein gives a complicated increase-decrease-increase-decrease signal (Kohout et al 2010; Mizutani et al 2022). Without understanding the mechanism behind the change in fluorescence, we hesitate to speculate on what the differences mean.

6. Authors should explain rationale for the idea that XI-VSP2 protein contributes to the

endogenous voltage dependent activity of increase of PI(3,4)P2 in Xenopus oocyte. It is not clear.

Based on our previous results studying both XI-VSP1 and XI-VSP2, we found that XI-VSP2 only dephosphorylates PI(3,4,5)P3 into PI(3,4)P2 and does not dephosphorylate PI(3,4)P2 into PI(4)P (Ratzen, 2019). Thus we are speculating that the dramatic rise in PI(3,4)P2 observed when testing enzymatically dead Ci-VSP is due to the endogenous XI-VSP2. If the XI-VSP1 was present, we would expect to see the dephosphorylation of PI(3,4)P2 into PI(4)P which we do not see even with the 10s long pulses. We have clarified our language in the text to reflect this explanation.

7. There seems to be a typo "H" in Fig. 3F.

We corrected the typo. Thank you for pointing it out!

January 11, 2024

Dr. Susy C. Kohout
Cooper Medical School of Rowan University
Biomedical Science
401 South Broadway
Camden, New Jersey 08103

Re: 202313467R1

Dear Susy,

Thank you for submitting your manuscript, entitled "S1 hydrophobic residues alter the voltage sensing phosphatase enzymatic function and voltage sensing" to JGP. Your manuscript has been seen by the same 3 reviewers, whose comments are appended below. You will see that although the reviewers found the manuscript was improved and a majority of their comments were addressed appropriately, they still have some lingering concerns that should be addressed prior to further consideration of the manuscript at JGP. In particular, please clarify the concerns related to the unexpected increase in PIP2 when using GFP-PLC-PH as a sensor in insulin-treated oocytes. In addition, a more precise evaluation of surface expression of the mutant enzymes may be necessary to ascertain that the observed shifts are due to intrinsic changes in voltage-dependence and not to altered protein surface expression. The reviewers recommend either a gating charge measurement or a surface biotinylation followed by pull-down and western blot. Other methods may also be suitable.

We would be pleased to receive a suitably revised manuscript that addresses these concerns, which will be re-reviewed, most likely by some or all of the original referees. Based on the scope of the requested changes, we typically anticipate that the revision process will take no longer than 6 months, however, we understand you may need additional time to work on your resubmission to JGP. We therefore ask that you simply keep us informed as to a realistic submission timeline that is appropriate for your particular circumstances. In addition, please do not hesitate to contact me (via the editorial office) if you feel that a discussion of the reviewers' and editors' comments would be helpful.

Please submit your revised manuscript via the link below along with a point-by-point letter that details your responses to the editors' and reviewers' comments, as well as a copy of the text with alterations highlighted (boldfaced or underlined). If the article is eventually accepted, it would include a 'revised date' as well as submitted and accepted dates. If we do not receive the revised manuscript within one year, we will regard the article as having been withdrawn. We would be willing to receive a revision of the manuscript at a later time, but the manuscript will then be treated as a new submission, with a new manuscript number.

Please pay particular attention to recent changes to our instructions to authors in the following sections: Data presentation, Blinding and randomization and Statistical analysis, under Materials and Methods, as shown here: <https://rupress.org/jgp/pages/submission-guidelines#prepare>. Re-review will be contingent on inclusion of the required information (including for data added during revision) and demonstration of the experimental reproducibility of the results. Also, to improve the reproducibility of published content, we have partnered with SciScore. Authors are prompted in eJP to copy and paste the Materials and Methods section of their manuscript for a SciScore assessment when submitting their revised manuscript. Authors are encouraged (not required) to further revise their Materials and Methods if the SciScore is below 4. More information can be found here: <https://rupress.org/jgp/pages/submission-guidelines#sciscore>

Please note, JGP now requires authors to submit Source Data used to generate figures containing gels and Western blots with all revised manuscripts (when applicable). This Source Data consists of fully uncropped and unprocessed images for each gel/blot displayed in the main and supplemental figures. If your paper includes cropped gel and/or blot images, please be sure to provide one Source Data file for each figure that contains gels and/or blots along with your revised manuscript files. File names for Source Data figures should be alphanumeric without any spaces or special characters (i.e., SourceDataF#, where F# refers to the associated main figure number or SourceDataFS# for those associated with Supplementary figures). The lanes of the gels/blots should be labeled as they are in the associated figure, the place where cropping was applied should be marked (with a box), and molecular weight/size standards should be labeled wherever possible. Source Data files will be made available to reviewers during evaluation of revised manuscripts and, if your paper is eventually published in JGP, the files will be directly linked to specific figures in the published article.

Source Data Figures should be provided as individual PDF files (one file per figure). Authors should endeavor to retain a minimum resolution of 300 dpi or pixels per inch. Please review our instructions for export from Photoshop, Illustrator, and PowerPoint here: <https://rupress.org/jgp/pages/submission-guidelines#revised>

When revising your manuscript, please be sure it is a double-spaced MS Word file and that it includes editable tables, if appropriate.

Please submit your revised manuscript via this link:
Link Not Available

Thank you for the opportunity to consider your manuscript.

Sincerely,

Crina Nimigean, Ph.D.
On behalf of Journal of General Physiology

Journal of General Physiology's mission is to publish mechanistic and quantitative molecular and cellular physiology of the highest quality; to provide a best-in-class author experience; and to nurture future generations of independent researchers.

Reviewer #1 (Comments to the Authors):

The revised manuscript titled "Impact of S1 hydrophobic residues on voltage sensing" by Rayaprolu and colleagues has mostly addressed my concerns. However, there are two major topics that require further discussion.

Major concerns:

On page 8, the authors describe the effect of mutation on the S1 segment on the phosphatase activity of Ci-VSP using fTAPP as sensor. When referring to the effect of the mutation L137A. The authors allude to the idea that the effect of the mutation is more profound on the 5'-phosphatase activity with respect to the 3'-phosphatase activity. Since these are "sequential reactions" as the product of the first one feeds the second reaction, the authors seem to tacitly rule out the possibility that the 5'-phosphatase activity is lower simply because of the limited supply of PI(4,5)P₂. The authors should consider explicitly addressing this point.

On page 9, the authors addressed their observation on an apparent increase in the concentration PI(4,5)P₂ as reported by the probe GFP-PLC-PH. Then, on page 11, the author provided further arguments supporting their observation. Notwithstanding the value of experimental data and the fact that the authors show an unambiguous increase in the signal, I found the explanation highly speculative and in need of further refinement. I would like to point out that I found that being able to track both the 3'- and 5'-phosphatase reaction with GFP-PLC-PH is simply remarkable. Yet, as pointed out before, this group of observations defies what is expected to happen with the concentration of PI(4,5)P₂ in the membrane. These experiments were performed with oocytes treated with insulin, aiming at increasing the concentration of PI(3,4,5)P₃. This fact is buried in the Methods section and in the discussion. So, explicitly reminding the readership that this treatment precedes the recording of fluorescence signal might be necessary. Moving forward, it would be great to see what happens with untreated oocytes. In the past, the Kohout lab has shown that the fluorescent signal from PLC decreases upon Ci-VSP activation, displaying a small initial increase in the signal. Although this has been shown in previous publications, showing what happens in the context of mutants would be informative. On the same topic, the initial increase in the concentration of PI(4,5)P₂ in untreated oocytes was attributed to the degradation of PI(3,4,5)P₃ into PI(4,5)P₂. So is the argument presented here. The question is, why doesn't the fPLC probe report that? If any, the signal should be much larger. Regardless of the affinities and other nuances inherent to the use of PH probes, the robust, unambiguous changes observed with the GFP-PLC-PH probe should also be reported by the fPLC probe. Does the different spatial distribution (subcellular localization) of the probes contribute to the difference in the signals? Is the affinity of the probe significantly different? I really regret to say that performing experiments with alternative methods for the detection of PIs (e.g.: ion channels as reporter) should be considered or at least discussed in detail.

Minor concerns:

1. On supplemental figure 2, the caption refers to Fig 6A and 6C. Shouldn't it be 5A and 5C?

Reviewer #2 (Comments to the Authors):

The manuscript has improved considerably with these revisions and all my major concerns have been addressed. There are few remaining points that need attention:

The new title "S1 hydrophobic residues alter the voltage sensing phosphatase enzymatic function and voltage sensing" is more informative than the previous title "Impact of S1 hydrophobic residues on voltage sensing". However, the use of the word "alter" implies a change from something else, e.g. from a reference condition or wild type, which can be confusing. It is suggested to use the word "modulate" instead. Alternatively, the authors could consider a compromise between the first and second version of the title, such as: "Modulation of voltage sensing and enzymatic activity by S1 hydrophobic residues in a voltage sensitive phosphatase".

In the revised abstract, the sentence "Co-immunoprecipitation demonstrated that none of the mutations broke the VSP dimer

indicating that the S1 impact could stem from intersubunit interactions" does not match what is proposed in Fig. 8, where both intersubunit and intrasubunit interactions are discussed.

A clearer rationale in support of the choice of S1 residues for this study is now provided in the results section. Still, the introduction should be more explicit about the difference between this work and previous studies, i.e. with a sentence like "while previous studies focused on S1 residues facing the core of the VSD, here we investigated residues predicted to face the other subunit in the VSP dimer".

Reviewer #3 (Comments to the Authors):

The aim of this study is to determine the influence of hydrophobic residues in the S1 of Ci-VSP on its voltage sensing properties and regulation of enzyme function. In the revised version, the Co-IP experiments and the statements of intermolecular effects have been moved to the Discussion section. Readability has been improved.

However, I still think the work is fragmented and the purpose has not been sufficiently achieved. The critical point is that the effects of mutations on expression levels have not been carefully considered. Its importance is very clear because even with a wild-type Ci-VSP, the apparent voltage dependency and kinetics of the phosphatase from the readouts of phosphatase probes are influenced by the amount of RNA to inject. This is because the output of the probe occurs through a complex "system" consisting of VSPs, substrates, endogenous suppliers and consumers, and the probe itself. For example, the lower the expression of VSP, the higher the apparent voltage dependence of phosphatase tends to be, which is likely due to the need to activate more phosphatase to achieve equivalent de-phosphorylation activity in low density situations. Therefore, careful attention should be paid to whether a mutation is really affecting the protein itself or the "system". The useful parameter for this is the net gating (or sensing) charge Q , which directly reflects the number of functional proteins in the membrane. Without this information, readers would not clearly understand the interpretations of the subsequent analysis using the phosphatase probes.

> We agree that expression will directly impact the outcome of the enzymatic activity assays. As such, for each cell used for an activity assay, we made sure the cells were expressing VSP by testing their VSD motions through VCF. In this context, the VCF serves the same purpose as the sensing current measurement. We clarified in the methods section that this control was conducted.

In theory, VCF data, when normalized by total fluorescence ($\Delta F/F$), reports protein movement, but not protein density. In real experimental situations, I agree that $\Delta F/F$ is dependent on expression levels because total fluorescence (F) is contaminated by the background membrane staining that does not show any response. Proper characterization of the background staining would be difficult. For this reason, the gating charge (Q) is essential to carefully examine the effects of mutations. The importance of this point is well recognized in other recent studies on VSP phosphatase activity including ref #10. I think it is prerequisite to report the Q -values ($[C]$ or $[C/F]$) of the cells expressing the mutants for publication as a JGP paper.

>Based on our previous results studying both XI-VSP1 and XI-VSP2, we found that XI-VSP2 only dephosphorylates PI(3,4,5)P3 into PI(3,4)P2 and does not dephosphorylate PI(3,4)P2 into PI(4)P (Ratzan, 2019). Thus we are speculating that the dramatic rise in PI(3,4)P2 observed when testing enzymatically dead Ci-VSP is due to the endogenous XI-VSP2. If the XI-VSP1 was present, we would expect to see the dephosphorylation of PI(3,4)P2 into PI(4)P which we do not see even with the 10s long pulses. We have clarified our language in the text to reflect this explanation.

Since phenomena alone are not sufficient for molecular identification (just as transient potassium currents are not always due to Shaker channels), we'd recommend avoiding a deterministic tone. In fact, my colleagues have tried antisense experiments in the past, but endogenous activity was not downregulated...

Rayaprolu et al- Response to Reviewer Comments, round 2

We thank all the reviewers for their constructive comments to help us improve this manuscript. Below are our responses (black) to specific comments (blue) and our first response to reviewers (blue, italics). We highlighted the revised text in green in the manuscript to make it easier to focus on the parts that changed.

Reviewer #1 (Comments to the Authors):

Major concerns:

On page 8, the authors describe the effect of mutation on the S1 segment on the phosphatase activity of Ci-VSP using fTAPP as sensor. When referring to the effect of the mutation L137A. The authors allude to the idea that the effect of the mutation is more profound on the 5'-phosphatase activity with respect to the 3'-phosphatase activity. Since these are "sequential reactions" as the product of the first one feeds the second reaction, the authors seem to tacitly rule out the possibility that the 5'-phosphatase activity is lower simply because of the limited supply of PI(4,5)P₂. The authors should consider explicitly addressing this point.

The fTAPP biosensor does not bind to PI(4,5)P₂ therefore a limited PI(4,5)P₂ supply will not be monitored. If the Reviewer meant PI(3,4)P₂ concentrations, then the 5-phosphatase reaction is observed "first" via the FRET increase and the 3-phosphatase reaction is observed "second". While the observations are sequential, they do not rule out that the reactions are occurring simultaneously. Our comments about the differences stem from a difference in kinetics compared to WT and a difference in the voltage dependence, again compared to WT. So if the PI(3,4)P₂ concentrations change the subsequent reactions, we expect a stronger impact on WT. Others (Grimm & Isacoff, 2016; Keum et al, 2016) have tackled the question of whether the reactions happen sequentially or simultaneously. Our results do not address this question therefore we believe it is out of the scope of this study.

On page 9, the authors addressed their observation on an apparent increase in the concentration PI(4,5)P₂ as reported by the probe GFP-PLC-PH. Then, on page 11, the author provided further arguments supporting their observation. Notwithstanding the value of experimental data and the fact that the authors show an unambiguous increase in the signal, I found the explanation highly speculative and in need of further refinement. I would like to point out that I found that being able to track both the 3'- and 5'-phosphatase reaction with GFP-PLC-PH is simply remarkable. Yet, as pointed out before, this group of observations defies what is expected to happen with the concentration of PI(4,5)P₂ in the membrane.

We appreciate the reviewers concerns. As you note, the increase in signal is unambiguous so the increase must be explained. Given that everyone who uses GFP-PLC-PH interprets the decrease in the fluorescence signal as a decrease in the PI(4,5)P₂ concentrations, we don't believe it is speculative to interpret the increase in the fluorescence signal as an increase in the PI(4,5)P₂ concentrations.

These experiments were performed with oocytes treated with insulin, aiming at increasing the concentration of PI(3,4,5)P₃. This fact is buried in the Methods section and in the discussion. So, explicitly reminding the readership that this treatment precedes the recording of fluorescence signal might be necessary.

We added a sentence in the results to ensure the readers know that the PI(3,4,5)P₃ concentrations have been increased through the insulin receptor pathway. Lines 319-321.

Moving forward, it would be great to see what happens with untreated oocytes. In the past, the Kohout lab has shown that the fluorescent signal from PLC decreases upon Ci-VSP activation, displaying a small initial increase in the signal. Although this has been shown in previous publications, showing what happens in the context of mutants would be informative.

The gPLC increases reported in this manuscript are similar to those we have reported in the past for wild type. In particular, Figure 2B from Castle et al shows both the fluorescence increase and decrease from gPLC and the dF/F values are in line with what we publish here for WT.

On the same topic, the initial increase in the concentration of PI(4,5)P₂ in untreated oocytes was attributed to the degradation of PI(3,4,5)P₃ into PI(4,5)P₂. So is the argument presented here. The question is, why doesn't the fPLC probe report that? If any, the signal should be much larger. Regardless of the affinities and other nuances inherent to the use of PH probes, the robust, unambiguous changes observed with the GFP-PLC-PH probe should also be reported by the fPLC probe. Does the different spatial distribution (subcellular localization) of the probes contribute to the difference in the signals? Is the affinity of the probe significantly different? I really regret to say that performing experiments with alternative methods for the detection of PIs (e.g.: ion channels as reporter) should be considered or at least discussed in detail.

The reviewer brings up interesting questions regarding the observed differences between gPLC and fPLC. We don't expect the absolute affinities between the two probes to be different since the same PH domain is used in both. However, the subcellular localization will impact the apparent affinity since being restricted to the membrane will change the on and off kinetics of fPLC compared to gPLC. Upon further reflection given the concerns of the reviewer, we realized that because fPLC is already close to the membrane, it could be close to saturated with the starting PI(4,5)P₂ concentrations. This could explain the extremely small signals we and others observe using fPLC. A similar logic may apply to ion channels. The Kir2.1, GIRK and KCNQ channels that we and others have used could be saturated with the starting PI(4,5)P₂ concentrations leaving no room for observing an increase in the PI(4,5)P₂ from the VSP reaction. This would be consistent with many other publications. We rewrote our discussion to add this extra information (Lines 521-537 & 547-549).

Minor concerns:

1. On supplemental figure 2, the caption refers to Fig 6A and 6C. Shouldn't it be 5A and 5C?

Corrected! Thank you for pointing out the typo.

Reviewer #2 (Comments to the Authors):

The manuscript has improved considerably with these revisions and all my major concerns have been addressed. There are few remaining points that need attention:

The new title "S1 hydrophobic residues alter the voltage sensing phosphatase enzymatic function and voltage sensing" is more informative than the previous title "Impact of S1 hydrophobic residues on voltage sensing". However, the use of the word "alter" implies a change from something else, e.g. from a reference condition or wild type, which can be confusing. It is suggested to use the word "modulate" instead. Alternatively, the authors could consider a compromise between the first and second version of the title, such as: "Modulation of voltage sensing and enzymatic activity by S1 hydrophobic residues in a voltage sensitive phosphatase".

We would be happy to use the suggested title, however, JGP has a character limit on titles so we cannot. We did manage to switch in the word "modulate" and stayed under the character limit.

In the revised abstract, the sentence "Co-immunoprecipitation demonstrated that none of the mutations broke the VSP dimer indicating that the S1 impact could stem from intersubunit interactions" does not match what is proposed in Fig. 8, where both intersubunit and intrasubunit interactions are discussed.

We added intrasubunit interactions in the abstract to better reflect our conclusions (Line 41).

A clearer rationale in support of the choice of S1 residues for this study is now provided in the results section. Still, the introduction should be more explicit about the difference between this work and previous studies, i.e. with a sentence like "while previous studies focused on S1 residues facing the core of the VSD, here we investigated residues predicted to face the other subunit in the VSP dimer".

We appreciate the reviewers efforts to improve our manuscript. The suggested sentence indicates that the previous studies specifically targeted S1 residues that face the core of the VSD. We do not know if

that is true or not. We feel that our extensively rewritten introduction provides the context the reader needs for our choice in S1 residues.

Reviewer #3 (Comments to the Authors):

The aim of this study is to determine the influence of hydrophobic residues in the S1 of Ci-VSP on its voltage sensing properties and regulation of enzyme function. In the revised version, the Co-IP experiments and the statements of intermolecular effects have been moved to the Discussion section. Readability has been improved.

However, I still think the work is fragmented and the purpose has not been sufficiently achieved. The critical point is that the effects of mutations on expression levels have not been carefully considered. Its importance is very clear because even with a wild-type Ci-VSP, the apparent voltage dependency and kinetics of the phosphatase from the readouts of phosphatase probes are influenced by the amount of RNA to inject. This is because the output of the probe occurs through a complex "system" consisting of VSPs, substrates, endogenous suppliers and consumers, and the probe itself. For example, the lower the expression of VSP, the higher the apparent voltage dependence of phosphatase tends to be, which is likely due to the need to activate more phosphatase to achieve equivalent de-phosphorylation activity in low density situations. Therefore, careful attention should be paid to whether a mutation is really affecting the protein itself or the "system". The useful parameter for this is the net gating (or sensing) charge Q , which directly reflects the number of functional proteins in the membrane. Without this information, readers would not clearly understand the interpretations of the subsequent analysis using the phosphatase probes.

> We agree that expression will directly impact the outcome of the enzymatic activity assays. As such, for each cell used for an activity assay, we made sure the cells were expressing VSP by testing their VSD motions through VCF. In this context, the VCF serves the same purpose as the sensing current measurement. We clarified in the methods section that this control was conducted.

In theory, VCF data, when normalized by total fluorescence ($\Delta F/F$), reports protein movement, but not protein density. In real experimental situations, I agree that $\Delta F/F$ is dependent on expression levels because total fluorescence (F) is contaminated by the background membrane staining that does not show any response. Proper characterization of the background staining would be difficult. For this reason, the gating charge (Q) is essential to carefully examine the effects of mutations. The importance of this point is well recognized in other recent studies on VSP phosphatase activity including ref #10. I think it is prerequisite to report the Q -values ($[C]$ or $[C/F]$) of the cells expressing the mutants for publication as a JGP paper.

Directly correlating the sensing charges for each cell used for the activity assays would require redoing all the activity experiments. Instead, to address the reviewers concerns about the mutant protein expression, we compiled all the VCF data for each cell that was used in the activity assays. The scatter plot of that data (added as a new supplemental figure 2) shows that the mutations had similar levels of VCF signal to those of both WT and the CS control. To test the statistical significance, we used a Welsh's t-test with a two-tailed distribution because the variances were not equal between the samples. All the mutants were statistically similar to WT. We also clarified that VCF was conducted for each cell included in the activity assays. (Lines 167-169, 417-419)

In addition, we conducted new experiments where the sensing current was measured in the same cell as VCF. We found that the F127A, I134A and S1-Q had statistically smaller amounts of charge when compared to WT, again using the Welsh's t-test to determine significance. In the same cells, the VCF were similar to WT for all except for S1-Q which showed a larger VCF signal. This difference is not surprising since the VCF captures more than just the sensing charge movements, but also captures a slower protein rearrangement. The smaller maximum sensing charges do not correlate with the observed shifts in phosphatase voltage dependence. For example, L137A gives similar maximal sensing charges when compared to WT while showing strongly shifted phosphatase activities and I134A gives lower maximal sensing charge compared to WT while showing almost no shifts in activity. To further allay the concerns of the reviewer, we note that when we have deliberately tested very low concentrations of Ci-

VSP (Rayaprolu et al 2018), we used Western blots to validate the protein expression and not VCF. This was because at very low concentrations of VSP on the membrane, VCF did not give a signal. Thus, the existence of the VCF signal indicates that the expression must be higher than the low expression conditions known to shift the phosphatase activity. We added this data to the supplemental (supplemental figure 2) and present the data in the methods and results (Lines 190-202, 265-268, 417-433).

>Based on our previous results studying both XI-VSP1 and XI-VSP2, we found that XI-VSP2 only dephosphorylates PI(3,4,5)P3 into PI(3,4)P2 and does not dephosphorylate PI(3,4)P2 into PI(4)P (Ratzen, 2019). Thus we are speculating that the dramatic rise in PI(3,4)P2 observed when testing enzymatically dead Ci-VSP is due to the endogenous XI-VSP2. If the XI-VSP1 was present, we would expect to see the dephosphorylation of PI(3,4)P2 into PI(4)P which we do not see even with the 10s long pulses. We have clarified our language in the text to reflect this explanation.

Since phenomena alone are not sufficient for molecular identification (just as transient potassium currents are not always due to Shaker channels), we'd recommend avoiding a deterministic tone. In fact, my colleagues have tried antisense experiments in the past, but endogenous activity was not downregulated...

We agree with the reviewer that our observations do not indicate a definitive molecular identification of the activity. We softened our language to indicate we are only suggesting the XI-VSP2 is responsible for the signal (Lines 333-335).

May 1, 2024

Dr. Susy C. Kohout
Cooper Medical School of Rowan University
Biomedical Science
401 South Broadway
Camden, New Jersey 08103

Re: 202313467R2

Dear Susy,

I am pleased to let you know that your manuscript, entitled "S1 hydrophobic residues modulate voltage sensing phosphatase enzymatic function and voltage sensing" is scientifically acceptable for publication in Journal of General Physiology. Formal acceptance will follow when it is modified in accordance with the referees' remarks and our editorial policies (please provide in the manuscript the duration of the inter-pulse intervals as referee 3 requests).

Please note items that need attention are listed at the bottom of this email (under 'manuscript formatting checklist') and on the attached marked-up pdf file. Please also be sure to include a copy of the text with alterations highlighted (boldfaced or underlined). Your manuscript should be a double-spaced MS Word file and include editable tables, if appropriate.

JGP requires a data availability statement for all research article submissions. These statements will be published in the article directly above the Acknowledgments. The statement should address all data underlying the research presented in the manuscript. Please visit the JGP instructions for authors for guidelines and examples of statements at <https://rupress.org/jgp/pages/editorial-policies#data-availability-statement>.

Lastly, JGP adds short captions to articles listed on our weekly newest article emails. If you haven't, please provide a short, ~40-word summary statement for the online JGP table of contents and alerts. This summary should describe the context and significance of the findings for a general readership and be placed on/near the title page.

Please submit your final files via this link:

Link Not Available

Thank you for choosing to publish your research in JGP and please feel free to contact me with any questions.

Sincerely,

Crina Nimigean, Ph.D.
On behalf of Journal of General Physiology

Journal of General Physiology's mission is to publish mechanistic and quantitative molecular and cellular physiology of the highest quality; to provide a best in class author experience; and to nurture future generations of independent researchers.

Manuscript formatting checklist:

- MS Word document of text needed (including editable tables)
- MS Word document of supplemental text needed (including figure legends and editable tables)
- Brief Statement describing supplementary information needed, if applicable (in subsection at end of Materials & Methods)
- Please include a data availability statement preceding the Acknowledgments section. Please see <https://rupress.org/jgp/pages/editorial-policies#data-availability-statement>
- References need to follow JGP style (This article has numbered citations in it, which is not the style we use). Please refer to our guidelines here: <https://rupress.org/jgp/pages/reference-guidelines>
- Figures created at sufficient resolution and in acceptable format (including supplemental if applicable). If working in Illustrator, we prefer .ai or .eps file format. If working in Photoshop please use 600dpi/1000dpi .tiff or .psd file format. Minimum resolution at estimated print size: Minimum resolution for all figures is 600 dpi. For figures that contain both photographs and line art or text, 600 dpi is highly recommended. Figures containing only black and white elements (line art, no color, and no gray) should be 1,000 dpi. Maximum figure size is 7 in wide x 9 in high (17.5 x 22.8 cm) at the correct resolution. <https://jgp.rupress.org/fig-vid-guidelines>
- Supplemental figures conforming to same guidelines as manuscript figures (noted above)
- Please be sure your Source Data Figure file is provided as an individual PDF files, *one file per figure*. (Authors should

endeavor to retain a minimum resolution of 300 dpi or pixels per inch. Please review our instructions for export from Photoshop, Illustrator, and PowerPoint here: <https://rupress.org/jgp/pages/submission-guidelines#revised>.)

- If images resemble one from a prior publications, the author must seek permissions (to reproduce or adapt) from the original publisher. [You can resubmit your paper while waiting to hear back from the original publisher but please keep us updated]

- All authors must complete a disclosure form prior to acceptance. A link to complete the form has been sent to all coauthors. Please provide the editorial office with updated email addresses if necessary

Reviewer #1 (Comments to the Authors):

My concerns have been addressed.

Reviewer #2 (Comments to the Authors):

My previous comments have been addressed appropriately. I have no further concerns.

Reviewer #3 (Comments to the Authors):

The manuscript has been improved and made clearer. Although the detailed mechanisms of the experimental results might be complex, the manuscript contains information that the reader may consider further. One additional suggestion is that if the authors have evidence that the inter-pulse intervals in Figures 2, 3, and 4 are long enough to allow the phosphatidylinositol environment (e.g., PIP2 levels) to recover to basal level, this should be clearly stated. Or, at least, the inter-pulse intervals used for these experiments should be provided in the manuscript.